# Single-cell profiling of HDAC inhibitor-induced EBV lytic heterogeneity defines abortive and refractory states in B lymphoblasts

Lauren E. Haynes[1], Ashley P. Barry[1], Micah A. Luftig [1,2]*

1 Department of Molecular Genetics and Microbiology, Duke University School of Medicine, Duke Center for Virology, Durham, North Carolina, United States of America, 2 Cancer and Stem Cell Biology, Duke-NUS Medical School, Singapore, Singapore

* micah.luftig@duke.edu

## Abstract

Epstein-Barr virus (EBV) is associated with multiple malignancies including Burkitt lymphoma (BL), Hodgkin's lymphomas, nasopharyngeal carcinomas (NPC), and gastric cancers. Canonically, EBV positive tumors display latent gene expression programs that are difficult to target pharmacologically. To overcome this hurdle, lytic reactivation therapies have been developed based on HDAC inhibition with limited mechanistic studies. We therefore characterized the impact of pan-HDAC inhibitor, panobinostat, and class I HDAC inhibitor, nanatinostat, on the growth, survival, and lytic reactivation of four EBV-positive cell lines: P3HR1-ZHT BL, Jijoye BL, IBL-1 immunoblastic lymphoma, and *de novo* infection derived lymphoblastoid cell lines (LCL). All lines were sensitive, enabling us to define ranges of sensitivity within which to use single cell approaches to assess early EBV lytic gene expression, cell cycle state, and apoptosis. We observed that each EBV-positive model of malignancy responded uniquely to the same HDAC inhibitors and that lytic reactivation was successful in only a small percentage of the cell population. To elucidate the potential role of host factors in preventing successful lytic reactivation, we performed single-cell RNA sequencing on the P3HR1-ZHT BL line treated with the HDAC inhibitor panobinostat. We observed that abortive lytic cells, or cells that do not successfully progress through the lytic cycle, upregulated genes downstream of NF-κB activity. Additionally, genes involved in immune signaling including the CD137/CD137L signaling axis, were upregulated in abortive lytic cells. Functional validation through a Cas9-RNP approach revealed that the CD137 receptor is indeed involved in preventing successful lytic reactivation. These data have important implications for how we approach oncolytic therapies for EBV-associated malignancies.

**Data availability statement:** Single cell RNAseq data have been uploaded to GEO under the following gene set: GSE319577. All other relevant data are in the manuscript and its Supporting information files.

**Funding:** This work was funded by NIH grant R01DE025994 and U01CA275306 (to M.A.L.), T32CA009111 (M.A.L.), Viracta Sponsored Research Agreement (M.A.L.) and F31DE033216 (to L.E.H.). The funders had no role in study design, data collection and analysis, decision to publish, or preparation of the manuscript.

**Competing interests:** The authors have declared that no competing interests exist.

## Author summary

Epstein-Barr virus (EBV) is an extremely prevalent human herpesvirus that is associated with a variety of cancers and autoimmune diseases. EBV establishes latent infection in the host and, under various circumstances, can reactivate the lytic cycle to produce more infectious particles. In the context of EBV-associated malignancies, the virus is most often maintained in a latent state, which makes it difficult to target with pharmaceuticals. To develop more viral targeted strategies, kick and kill regimens have been investigated. This therapy involves reactivating the virus with an HDAC inhibitor followed by treatment with an antiviral drug. It is well established that reactivating EBV with pharmaceuticals is often inefficient and leads to heterogeneous responses, including an abortive lytic trajectory. To better characterize the overall effect of two classes of HDAC inhibitors in various models of EBV-associated malignancies, we utilized single-cell techniques to capture various responses to stimuli. Consistent with prior studies, HDAC inhibition led to both successful and abortive lytic populations. Single-cell RNA sequencing provided evidence of upregulated immune signaling pathways in this abortive lytic population. Downstream functional validation revealed that CD137 signaling is important for this abortive lytic phenotype. This study provides in depth characterization of lytic reactivation with a biologically relevant stimulus.

## Introduction

Epstein-Barr virus (EBV) is an extremely pervasive γ-herpesvirus, infecting approximately 95% of adults worldwide [1]. Primary infection is usually asymptomatic and occurs in the first decade of life. If infection occurs during adolescence, it can lead to infectious mononucleosis. Following primary infection, EBV establishes and maintains latency in a small compartment of memory B cells for life [2]. Latent infection can promote the development of a variety of malignancies including diffuse large B cell lymphomas (DLBCL), NK/T cell lymphomas, nasopharyngeal carcinoma, gastric cancer, Burkitt lymphoma (BL), and Hodgkin's lymphoma [1,3,4]. In most of these EBV associated cancers the virus remains latent and expresses very few genes, making it difficult to kill infected tumor cells with virus-specific therapeutics. Like other herpesviruses, EBV has a biphasic life cycle which includes both latency and lytic replication. *In vivo*, EBV-infected B cells reactivate when they undergo plasma cell differentiation or cellular stress [5]. This reactivation leads to the production of infectious viral particles that infect epithelial cells in the oral mucosa and are shed through saliva [6].

To better target EBV-positive malignancies, viral reactivation therapies have been developed, and some have even entered clinical trials [7–14]. These therapies involve treatment with a compound that reactivates the EBV lytic cycle followed by treatment with a herpesvirus specific antiviral drug, often ganciclovir (GCV) [15,16]. Previous investigations into EBV lytic reactivation have found various compounds

that can lead to reactivation both *in vitro* and *in vivo*. In earlier studies, γ-irradiation, histone deacetylase (HDAC) inhibitors like sodium butyrate and arginine butyrate, and the phorbol ester TPA/PMA were found to successfully reactivate EBV and sensitize cell lines or tumors to antiviral pro-drugs [8,14]. More recent investigations have found that tetrahydrocarboline derivatives are also effective at reactivating EBV in multiple infected cell lines [11]. Barriers to translating this strategy in the clinic have included tumor/cell type-specific responses, inefficient reactivation, and general toxicity.

Of the compounds mentioned above, HDAC inhibitors have been widely studied as anticancer drugs and have moved the furthest through clinical trials [17,18]. The HDAC inhibitors vorinostat, romidepsin, and belinostat have been approved for the treatment of certain T cell lymphomas and panobinostat has been studied in clinical trials for the treatment of multiple myeloma [17–19]. HDACs have been classified based on sequence homology. Class I HDACs include HDAC1, -2, -3, and -8. Class II is comprised of HDAC4, -5, -6, -7, -9, and -10 while class IV includes only HDAC11 [20]. Pan-HDAC inhibitors inhibit all classes of HDACs while other inhibitors target specific classes. Functionally, HDACs remove acetyl groups from histones which typically leads to chromatin condensation and transcriptionally repressed genes [17,21]. Therefore, treatment with HDAC inhibitors leads to hyper-acetylation on histones since acetyl groups are unable to be removed [20,22]. Other proteins including chromatin remodeling proteins, DNA binding transcription factors, structural proteins, and signaling mediators can also have their activity altered by changes in acetylation [17]. This activity can often lead to cell cycle arrest and DNA damage [23].

While HDAC inhibitors and their effects have been studied for use in combination with traditional anti-cancer drugs and radiation, they are also studied for use in EBV lytic reactivation therapies [16,24,25]. Recently, a class I HDAC inhibitor called nanatinostat has moved through a phase I/IIb clinical trial for treatment of multiple EBV-associated lymphomas including DLBCL, Hodgkin's lymphomas, and NK/T cell lymphomas [7,26]. In EBV infected cells, reactivation, or the switch from latency to lytic replication is tightly regulated [27]. The lytic cycle is governed by a temporal cascade of gene expression where the immediate-early (IE) gene products BZLF1 (Z, Zta) and BRLF1 (R, Rta) function as transcription factors for the early genes which in turn lead to the expression of the late genes [28–30]. Importantly, expression of late genes is contingent upon successful viral DNA replication and the formation of the viral pre-initiation complex (vPIC) [31–33]. Early gene products consist of the viral DNA polymerase and viral kinases, while late gene products include tegument and capsid proteins as well as glycoproteins [28]. For the successful reactivation of the lytic cycle, HDAC inhibitors must be able to promote expression of one or both IE genes. Additionally, this regimen relies on successful progression through the lytic cycle since downstream antivirals require the presence of early lytic stage viral kinases [34].

The precise mechanism by which HDAC inhibitors promote lytic reactivation is currently unknown. In the latent phase, EBV exists as an episome bound by histone proteins and tethered to the host genome [35]. This compaction of viral DNA helps promote latency and it is thought that HDAC inhibitor mediated hyperacetylation of IE lytic gene promoters may be important for the subsequent activation of these genes [36]. Later it was observed that protein kinase C delta (PKCδ) was required for HDAC inhibitor dependent EBV reactivation, and it has been hypothesized that PKCδ activity leads to the release of the HDAC2 repressor from the Z promoter [37,38]. More recently it has been shown that CTCF binding sites on viral genes are important for the switch from latency to lytic reactivation in multiple EBV-positive backgrounds [39]. These previous studies highlight that genome architecture and epigenetic modifications play a large role in successful viral lytic reactivation.

In this study we characterized the effects of the class I HDAC inhibitor nanatinostat and the pan-HDAC inhibitor panobinostat on four distinct EBV-positive lymphoblast cell lines: two related BL cell lines (Jijoye and its subclone, P3HR1 [40], modified to inducibly express BZLF1), an AIDS-associated immunoblastic lymphoma line (IBL-1), and a freshly-derived lymphoblastoid cell line (LCL), which models the precursor state of activated B cell diffuse large B cell lymphomas (ABC-DLBCL). These cell backgrounds were chosen as they are established EBV-positive models of various malignancies, although they are not models of current clinical studies. Nanatinostat has shown clinical promise in treating EBV-positive lymphomas, particularly in non-Hodgkin lymphomas and T/NK cell lymphomas [7,26,41]. However, there has

been little mechanistic work interrogating nanatinostat activity in EBV-positive reactivation models. Panobinostat was also chosen to perform follow up studies as panobinostat and nanatinostat have distinct targets from each other and treatment may lead to differential responses. Indeed, it has previously been shown that while various classes of HDAC inhibitors can lead to reactivation, heterogeneous responses in EBV-positive cells have been observed [9,42]. In our recent work, we also observed unique cell states and heterogeneous responses during lytic reactivation in an inducible model of reactivation [43].

In addition to characterizing the overall effect of these HDAC inhibitors in various models of EBV malignancy, we were further interested in characterizing the heterogeneous fates of cells following treatment. To address these questions, we utilized multiple single-cell approaches including RNA Flow FISH and scRNA-seq following treatment with HDAC inhibitors. While we observed the expected cell cycle arrest and cell death following treatment with these stimuli, this did not prevent a population of cells from undergoing successful lytic reactivation. We also observed that different malignancy backgrounds had unique responses to HDAC inhibition. Our RNA Flow FISH and scRNA-seq experiments highlighted that there is incomplete progression through the lytic cycle, at least in part governed by the expression of host factors involved in immune signaling including CD137.

## Materials and methods

### Cell lines

Cell lines used in this study include: Jijoye (purchased from Duke University Cell Culture Facility; RRID:CVCL_1317), IBL1 (kind gift of E. Cesarman, Weill Cornell; RRID:CVCL_9638), BL41 (RRID:CVCL_1087) and BL41-B958 (RRID:CVCL_C5V2) obtained from George Mosialos (Aristotle University, Thessaloniki, Greece), and lymphoblastoid cell lines (LCLs) generated in our lab. LCLs are derived from PBMCs isolated from buffy coats (Gulf Coast Regional Blood Center) via Ficoll gradient, infected with B95.8 EBV, and allowed to transform over the course of 5 weeks. P3HR1-ZHT was also used (kind gift of E. Kieff, Harvard Medical School). P3HR1-ZHT is an EBV-positive Burkitt lymphoma (RRID:CVCL_2676) that contains the EBV IE gene BZLF1 fused to a 4-hydroxytamoxifen (4HT) dependent mutant estrogen receptor binding domain [44]. Treatment with 4HT (H7904, Millipore Sigma) leads to effective reactivation of the EBV lytic cycle. All cell lines were cultured in RPMI media (11878093, Gibco) supplemented with 10% fetal bovine serum (FBS) and maintained at 37°C with 5% $CO_2$.

### $CC_{50}$ (cytotoxicity concentration) curves

Cells were plated in biological triplicate at 400,000 cells/mL and treated with 1–10,000 nM of the HDAC inhibitors panobinostat and nanatinostat at half log-fold dilutions. Cells were collected at 48h post treatment and incubated with Cell-Titer Glo (Promega) for 10 minutes at room temperature. Luminescence was detected with Promega plate reader as a proxy for cell viability. Analysis was completed in Prism (Graphpad). First, concentrations were transformed into log of dose ($x = \log(x)$). Nonlinear regression (curve fit) was then performed (log(inhibitor) vs normalized response-variable slope). $CC_{50}$ values were determined based on the nonlinear fit of the transformed concentration values. The x-axis of each graph was changed to anti-log for visualization purposes and to show the concentrations used.

### Cell cycle analysis

Cells were plated in biological triplicates at 400,000 cells/mL and treated with multiple concentrations of the pan-HDAC inhibitor panobinostat and the Class I HDAC inhibitor nanatinostat for 24h and 48h. The BD Pharmingen BrdU kit (BD Biosciences) was used to determine cell cycle stage according to kit protocol. Briefly, cells were collected and incubated with 1 mM BrdU for 2h at 37°C. Cells were washed with FACS buffer (PBS with 2% FBS) and then fixed in cytofix/cytoperm buffer for 30 minutes at room temperature. Cells were washed with Perm/Wash buffer and incubated with cytoperm

permeabilization buffer plus at 4°C for 10 minutes, washed, and re-fixed for 5 minutes at room temperature. Cells were treated with DNase for 1 hour at 37°C, washed and incubated with 4 µL mouse anti-BMRF1 (EA-D) (Santa Cruz Cat # 69679) for 30 minutes at room temperature. Cells were washed and incubated in anti-mouse A647 (Thermofisher Scientific) for 15 minutes at room temperature and then stained for BrdU with a FITC anti-BrdU antibody for 20 minutes. Finally, cells were incubated with 7-AAD for 15 minutes at room temperature and analyzed on a Canto II Cytometer (BD biosciences). Analysis was completed on FlowJo software (BD biosciences). Statistics were performed using Student's t-test.

### Intracellular flow cytometry

Cells were plated in biological triplicate at 400,000 cells/mL and treated with multiple concentrations of the pan-HDAC inhibitor panobinostat and the Class I HDAC inhibitor nanatinostat for 48h. 500,000 cells were harvested and washed with 1X PBS. Cells were stained with 1 µL Zombie Violet (Biolegend) for a final concentration of 1:200 for 15 minutes at room temperature and washed with FACS buffer. Cells were then washed twice with 1X PBS and fixed with BD Cytofix/Cytoperm buffer (BD biosciences) for 20 minutes at 4°C. Cells were pelleted and washed with Perm/Wash buffer. Cells were incubated with 20 µL FITC-Cleaved Caspase-3 (BD) and 4 µL mouse anti-BMRF1 (EA-D) (Santa Cruz Cat # 69679) for 30 minutes at room temperature. Cells were washed and incubated in anti-mouse A647 (Thermofisher Scientific) for 15 minutes at room temperature and subsequently analyzed on a Canto II Cytometer (BD biosciences). Analysis was completed on FlowJo software (BD biosciences). Statistics were performed using Student's t-test.

### RNA flow FISH

Cells were plated at 400,000 cells/mL and treated with DMSO, 200 nM nanatinostat, or 50 nM panobinostat. Cells were harvested at 24h and 48h post treatment with HDAC inhibitors. RNA Prime Flow kit and protocol from ThermoFisher was followed with no adjustments. Briefly, cells were washed, fixed, and permeabilized. Cells were then incubated with target probes for 2h in a 40°C water bath. Target probes used in this paper included the EBV mRNAs BZLF1, BGLF4, and BLLF1 and the host mRNAs STAT1, STAT3, and MYC. Cells were washed and stored overnight at 4°C and then incubated with a Pre-amplification buffer for 1.5h in a 40°C water bath followed by a 1.5h incubation in amplification buffer. Cells were then incubated in label probes for 1 hour in a 40°C water bath, washed with FACS buffer and subsequently analyzed on a Cytek Aurora cytometer. Spectral flow unmixing was completed using the SpectroFlo (Cytek) software and downstream analysis was performed on FlowJo (BD biosceinces). Experiments were performed in technical triplicate or duplicate. Statistics were performed using one-way ANOVA.

### Single cell library prep and analysis

P3HR1-ZHT cells were plated at 400,000 cells/mL and treated with 50 nM panobinostat for 48h. Cells were collected for library preparation and the viability at the time of collection was 73%. Cells were resuspended in the proper concentration to collect ~10,000 cells for GEM generation. Single-cell transcriptome was captured and reverse transcribed into cDNA libraries using the 10X Genomics Chromium Next GEM Single Cell 3' gene expression kit v3.1 chemistry and Chromium microfluidics controller according to the 10X protocol. Libraries were sequenced on a Nova-Seq6000 with 50 base pair paired-end reads at a targeted sequencing depth of 50,000 reads per cell. Cell Ranger (10x Genomics) was used for read alignment and quality control. Base calls were assembled into fastq reads and were mapped to a concatenated human and EBV genome package (hg38 + NC_09334 [type 2 EBV]). Single-cell gene expression matrices were generated to create a Seurat object in R. The untreated time point was prepared separately following the same protocol listed above and was used in our previous work [43].

Our Seurat object was subject to quality control filters and cells were included if: gene expression was identified in a minimum of 3 cells, mitochondrial genes accounted for < 25% of all transcripts, there were a minimum of 200 unique genes expressed, and there were < 65,000 total transcripts (to excluded non-singlets). Raw count data was normalized

and scaled using the *NormalizeData* and *ScaleData* functions in the Seurat package. The data from both time points (untreated and 48h post treatment) were integrated into a single Seurat object using the *FindIntegratedAnchors* and *IntegrateData* functions. Expression data was dimensionally reduced using principal component analysis and the *RunPCA* function. The first 30 principal components were used for downstream UMAP generation using the *FindNeighbors* and *RunUMAP* functions. Cell clusters were identified using the *FindClusters* function for phenotype identification and downstream analysis. Finally, to correct technical read dropout we imputed our dataset using adaptive low-rank approximation (ALRA) of the RNA count matrix [45]. All data (except for CellChat) in this study was generated from imputed read data.

### Cell chat analysis

All Cell Chat analysis was completed using the CellChat package in R [46]. A cellchat object was created using our Seurat object as a data frame. To increase the chances of identifying functional hits, we started our analysis with the un-imputed single cell data (no ALRA). We used CellChat's curated list of 1,939 validated human signaling interactions to identify overexpressed genes and overexpressed interactions from our dataset. The *computeCommunProb* function was used to compute communication probability and infer cellular communication networks and filtered out hits if there were less than 10 cells identified in certain cell groups. We then used the *computeCommunProbPathway* function to infer cell-cell communication at a signaling pathway level.

### Extracellular staining for abortive lytic markers

Cells were plated in biological triplicate at 400,000 cells/mL and treated with DMSO, 50 nM panobinostat, or 50 nM panobinostat with 1 μg/mL phosphonoacetic acid (PAA) for 48h. Cells were harvested and washed with FACS buffer (PBS with 5% FBS). Cells were stained with 1 μg fluorophore-conjugated 72A1 antibody (gp350) and 5 μL of anti-human CD137 (Biolegend 309804) or 5 μL anti-human VCAM-1 (ThermoFisher A15439) and incubated for 30 minutes at room temperature in the dark. Cells were washed with FACS buffer and analyzed on a canto II cytometer (BD biosciences). For CD137L staining, cells were collected, washed with FACS buffer, and stained with 5 μL anti-human CD137L (Biolegend 311508) and fluorophore-conjugated 72A1 or anti-human CD137. Cells were incubated for 30 minutes at 4°C in the dark. Cells were washed with FACS buffer and fixed with BD Cytofix/Cytoperm buffer (BD biosciences) for 15 minutes at 4°C in the dark. Cells were washed with FACS buffer and analyzed on a canto II cytometer (BD biosciences). All downstream data analysis was completed in FlowJo software (BD biosciences). Statistics were performed using one-way ANOVA.

### CRISPR Cas9-RNP approach for targeting abortive lytic host markers

Cas9-RNP complexes were prepared by combining 10 pmol Cas9 (TrueCas9 V2 ThermoFisher), 30 pmol of sgRNA (Synthego), and resuspension buffer (Buffer R ThermoFisher). For each experimental knockout, up to three sgRNAs were used targeting a 150-bp spanning region of the gene of interest. A sgRNA targeting the cell surface marker CD46 was added to each experimental condition to serve as a proxy for successful transfection. The sgRNAs used in this study are listed in S2 Table. The Cas9-RNP complexes were combined with 500,000 P3HR1-ZHT cells and subsequently transfected using the 10 μL Neon electroporation system. Electroporation parameters used in this study include: 1 pulse, 1350mV, 40ms. Immediately after transfection, cells were plated in RPMI media (11878093, Gibco) supplemented with 15% fetal bovine serum (FBS) and left to recover for 72h at 37°C with 5% $CO_2$.

## Results

### EBV-positive cell lines display differential sensitivity to HDAC inhibition

We first tested the responsiveness of four EBV-positive cell lines to the pan-HDAC inhibitor panobinostat and the class I HDAC inhibitor nanatinostat. We chose EBV-positive cell lines representative of pre-malignant (LCL) and malignant

(BL and IBL) states and determined their cytotoxicity concentration ($CC_{50}$) at 48h following treatment. The $CC_{50}$ for panobinostat in the BL cell line P3HR1-ZHT was 13.7 nM while the $CC_{50}$ for nanatinostat was 122.9 nM (Fig 1A). Similarly, the $CC_{50}$ for panobinostat in the BL cell line Jijoye was 15.5 nM while the $CC_{50}$ for nanatinostat was 135.8 nM (Fig 1B). The AIDS immunoblastic lymphoma line IBL1 and the LCL exhibited similar responses to both HDAC inhibitors and comparable $CC_{50}$ concentrations. IBL1 had a $CC_{50}$ of 8.4 nM for panobinostat and a $CC_{50}$ concentration of 61.1 nM for nanatinostat (Fig 1C) while the LCL had a $CC_{50}$ concentration of 9.5 nM for panobinostat and 90 nM for nanatinostat (Fig 1D). These $CC_{50}$ values highlight that different malignancy backgrounds may exhibit differential sensitivity to HDAC inhibitor treatment.

To determine whether the effect of HDAC inhibition was dependent on EBV-status, we treated the EBV-negative BL line BL41 and the matched EBV-infected line BL41-B958 with nanatinostat and panobinostat for 48h (Fig 1E, 1F). Both cell lines were sensitive to HDAC inhibition and exhibited similar $CC_{50}$ values following treatment. These data show that HDAC inhibition induced cell death is agnostic to the presence of EBV. Across all cell lines tested and regardless of EBV status, nanatinostat treatment was less toxic than panobinostat. This is likely because panobinostat has a broader range of targets compared to nanatinostat. We used these $CC_{50}$ values as a guide for working concentrations in subsequent experiments to strike a balance between cell cycle arrest, cell death, and potential lytic reactivation. To provide a wider window of treatment for our future experiments, we also calculated the $CC_{30}$ and $CC_{70}$ concentrations for nanatinostat and panobinostat (S1 Table).

## EBV-positive cell lines have a heterogeneous response to HDAC inhibitors at the protein level

To characterize the balance between cell cycle arrest and potential lytic reactivation, we treated each cell line with increasing concentrations of nanatinostat or panobinostat for 48h and stained for BrdU, 7AAD, and the EBV early lytic protein BMRF1. This staining method allowed us to capture viral DNA replication and differentiate between cells that respond to treatment and start the process of lytic reactivation versus cells that do not respond. In addition, this technique allowed us to visualize single-cell dynamics within a population.

In the P3HR1-ZHT background, most of the BMRF1+ cells were in S phase even though treatment led to cell cycle arrest (Fig 2A, red cells). BMRF1+ cells in S phase indicate that this labeling was viral DNA replication. All concentrations of nanatinostat or panobinostat tested led to significantly increased levels of BMRF1 (Fig 2B). Moreover, the 200 nM nanatinostat and 50 nM panobinostat treatments led to similar levels of reactivation in this line. In the Jijoye line, BMRF1+ cells were present in either S phase or G1 (Fig 2C, red cells). Each concentration of HDAC inhibitor tested led to significantly increased lytic reactivation, although overall reactivation was lower compared to the P3HR1-ZHT line (Fig 2D). The higher concentrations of panobinostat and nanatinostat led to similar levels of reactivation in this line. In the IBL1 line, we observed a decrease in the percentage of cells in S phase at lower concentrations of HDAC inhibitors (Fig 2E). Additionally, there was no appreciable BMRF1 detected (Fig 2F). In the LCL, most cells remained arrested in G1 following treatment with both HDAC inhibitors (Fig 2G). Moreover, by 48h post treatment we did not detect BMRF1 in this cell background (Fig 2H). Overall, this staining pattern highlights that while HDAC inhibitors do lead to a level of cell cycle arrest in the P3HR1-ZHT and Jijoye background, this does not prevent a certain population of cells from reactivating. Across all concentrations of HDAC inhibitor tested, P3HR1-ZHT had the least overall percentage of cells in Sub G1 (S1 Fig). Conversely, there was more cell cycle arrest (G1) in the P3HR1-ZHT line compared to the Jijoye line (S1 Fig). IBL1 and the LCL displayed similar patterns of arrest across concentrations tested with limited cells in the Sub G1 population, except for the highest concentrations tested (S1 Fig).

In the BL lines, lytic reactivation at the protein level was dose dependent, with higher concentrations leading to higher levels of protein expression (S2 Fig). In the IBL1 line, BMRF1 protein was not detected above background across all concentrations tested at 48h post treatment (S2C Fig). Similarly, in the LCL there was limited BMRF1 staining above background with any concentration of HDAC inhibitor after 48h (S2D Fig). As an additional control for lytic reactivation, we treated P3HR1-ZHT cells with 4-hydroxytamoxifen (4HT) and stained for the cell cycle and BMRF1. We visualized both

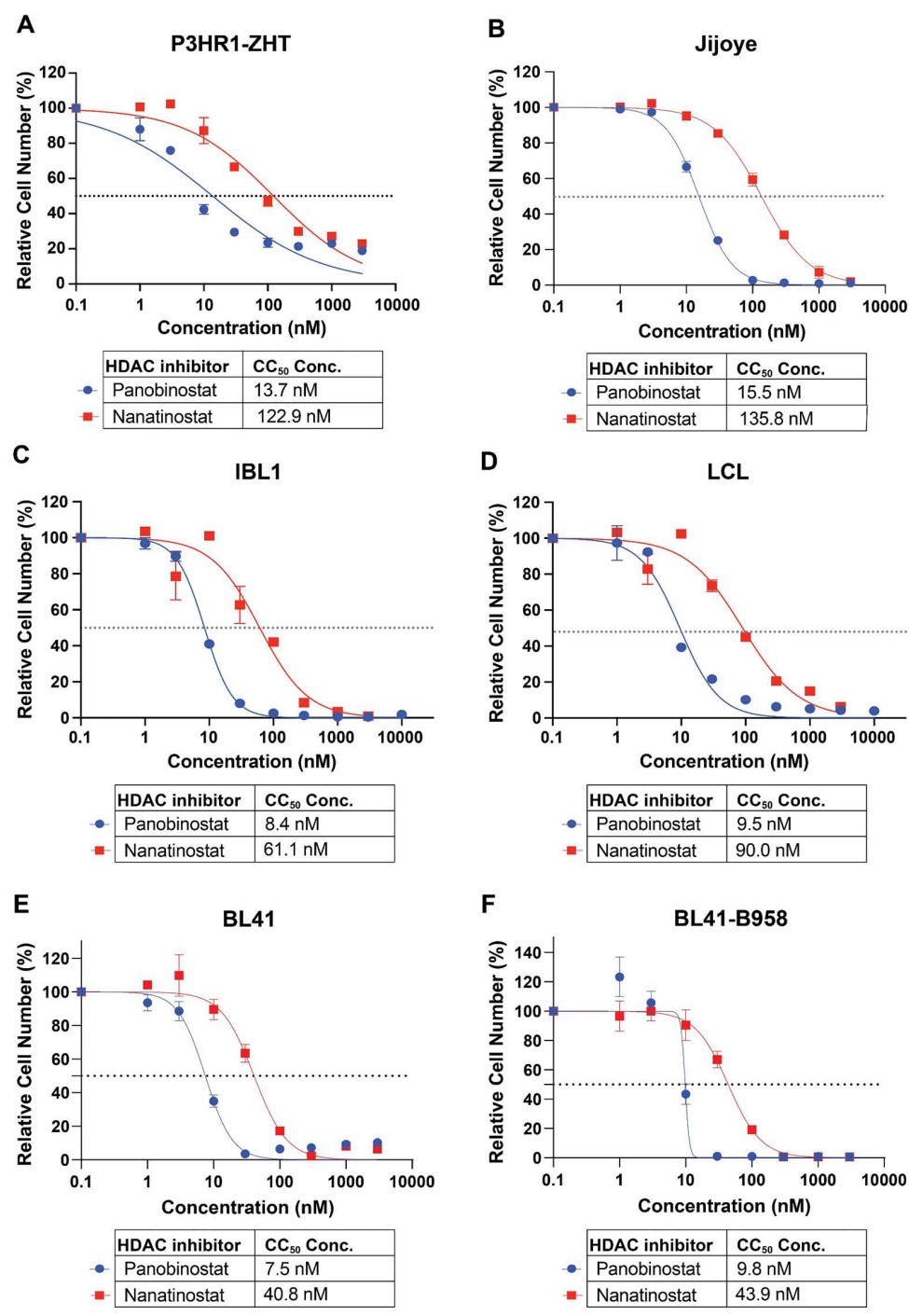

**Fig 1. HDAC inhibitors suppress EBV+B cell proliferation.** (A) CC$_{50}$ curves of P3HR1-ZHT, (B) Jijoye, (C) IBL1, (D) an LCL, (E) BL41, (*F*) and BL41-B958 treated with the pan HDAC inhibitor panobinostat and the class I inhibitor nanatinostat. All cell lines were treated for 48h, and live cells were measured with CellTiter Glo and normalized to DMSO treatment.

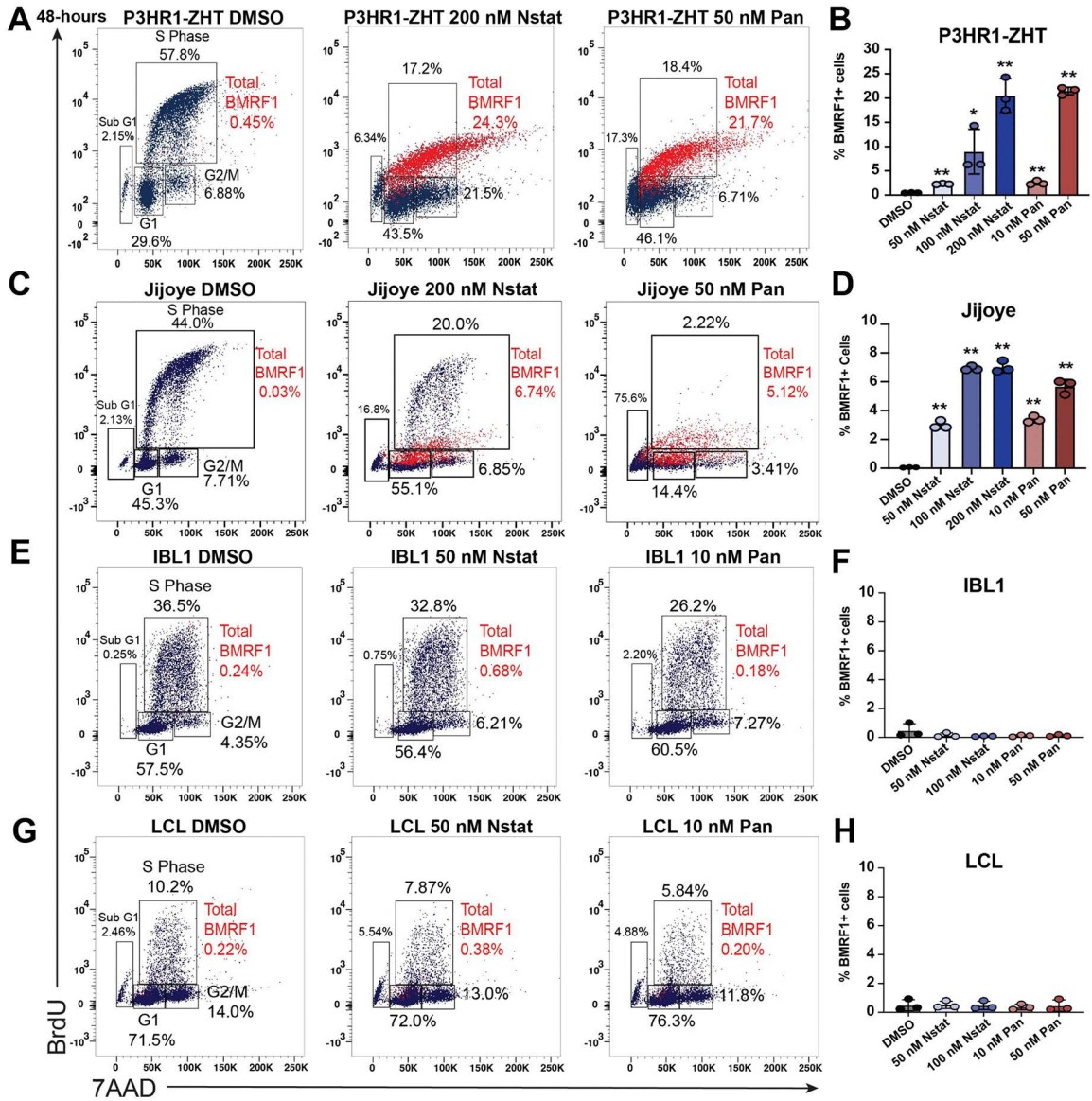

**Fig 2. BL cell lines are most responsive to HDAC inhibitor treatment regardless of cell cycle arrest.** (A) P3HR1-ZHT cells were treated with increasing concentrations of panobinostat and nanatinostat for 48h. Cells were collected and stained for the EBV lytic product BMRF1, BrdU, and 7AAD. (B) Percentage of BMRF1+P3HR1-ZHT cells after treatment with three concentrations of nanatinostat and two concentrations of panobinostat. Data points from experiment in panel A. (C) Jijoye cells were treated with increasing concentrations of panobinostat and nanatinostat for 48h. Cells were collected and stained for the EBV lytic product BMRF1, BrdU, and 7AAD. (D) Percentage of BMRF1+Jijoye cells after treatment with three concentrations of nanatinostat and two concentrations of panobinostat. Data points from experiment in panel C. (E) IBL1 cells were treated with increasing concentrations of panobinostat and nanatinostat for 48h. Cells were collected and stained for the EBV lytic product BMRF1, BrdU, and 7AAD. (F) Percentage of BMRF1+IBL1 cells after treatment with two concentrations of nanatinostat and two concentrations of panobinostat. Data points from experiment in panel E. (G) LCLs were treated with increasing concentrations of panobinostat and nanatinostat for 48h. Cells were collected and stained for the EBV lytic product BMRF1, BrdU, and 7AAD. (H) Percentage of BMRF1+LCLs cells after treatment with two concentrations of nanatinostat and two concentrations of panobinostat. Data points from experiment in panel G. For panels B,D,F,H n = 3 *p < 0.05, **p < 0.01.

BMRF1+ and BMRF1- cells in S phase as two distinct populations (S3 Fig). This inducible method of reactivation does not lead to the significant level of cycle arrest that we observed following HDAC inhibition.

**Visualization of reactivation continuum in EBV-positive malignancy models following HDAC inhibition**

To further probe dynamics of reactivation and elucidate the trajectory of cells, we examined the level of cell death, BMRF1 protein expression, and cleaved caspase-3 following 48h of HDAC inhibitor treatment. We chose to examine cleaved caspase-3 as a marker of early apoptosis which allowed us to visualize which cells might be poised to undergo apoptosis that we would otherwise not have been able to see by live/dead staining. In the P3HR1-ZHT line, we observed distinct fate outcomes captured at 48h post treatment. Following treatment, cells will either undergo cell death (BMRF1-, Zombie Violet+), reactivate the lytic cycle (BMRF1+, Zombie Violet-), or remain refractory to treatment (BMRF1-, Zombie Violet-). In P3HR1-ZHT cells there were also cases of double positivity for lytic protein expression and cleaved caspase-3, particularly following treatment with panobinostat (Fig 3A). Any concentration of HDAC inhibitor led to significantly increased cell death in this cell line in a dose dependent manner, with the highest level of cell death following treatment with 50 nM of panobinostat (Fig 3B). Similarly, cleaved caspase-3 increased in a dose dependent manner, with the highest level observed in 50 nM panobinostat (Figs 3B, S4A). While cell death did significantly increase after each treatment in this cell line, no concentration of HDAC inhibitor led to more than ~10% of cells positive for Zombie Violet (Fig 3B). In the other BL line Jijoye, we observed a more exaggerated dose dependent increase in the amount of cell death and cleaved caspase-3 following HDAC inhibition (Figs 3D, S4B). While the highest concentrations did lead to the highest level of cell death, they also led to the most overall lytic reactivation (Figs 3C, 2D). Additionally, we observed double positive cells for cleaved caspase-3 and BMRF1 following treatment with nanatinostat, similar to what we observed in the P3HR1-ZHT line (Fig 3C).

In the IBL1 line, we again detected limited BMRF1 expression after treatment with HDAC inhibitors for 48h (Fig 3E). Overall, most IBL1 cells exhibited minimal response to treatment. The highest level of cell death was observed following treatment with 250 nM nanatinostat and 50 nM panobinostat (Fig 3F). Furthermore, most cleaved caspase-3 was detected in the highest concentration of nanatinostat and panobinostat (Figs 3F, S4C). The LCL also exhibited minimal response to treatment and increased cell death at higher concentrations of HDAC inhibitor, particularly 50 nM panobinostat (Fig 3G, 3H). Cleaved caspase-3 similarly increased in a dose dependent manner, with the highest level observed following treatment with 50 nM panobinostat (Figs 3H, S4D). In all cell lines, cleaved caspase-3 was observed in the refractory population (BMRF1-, Zombie Violet-) indicating that this population is pre-apoptotic (Fig 3A, 3C, 3E, 3G red cells). Overall, cleaved caspase-3 staining was more pronounced in the panobinostat treatment compared to nanatinostat treatment due to the more toxic nature of this HDAC inhibitor.

This staining pattern revealed that a large proportion of cells within a population are refractory to treatment. This was most apparent in the IBL1 line and the LCL, where >90% cells were both BMRF1-negative and Zombie-negative (Fig 3E, 3G). In the IBL1 treated cells, we observed that treatment with lower doses of nanatinostat led to minimal cell death compared to that of treatment with panobinostat for 48h (Fig 3F). In an effort to increase the level of lytic reactivation, we treated IBL1 with 50 nM and 100 nM nanatinostat for 72h. The additional treatment time led to an increase in the percentage of cells that reactivate and express BMRF1 while maintaining the same percentage of dead cells and cleaved caspase-3 (Figs 3I, S5C). We extended the treatment time and concentration of nanatinostat for the LCL but did not observe an increase in the percentage of cells that express BMRF1 (S5A, S5B Fig). Extending the treatment time of P3HR1-ZHT or Jijoye cells to 72h did not lead to additional BMRF1+ cells, indicating that peak reactivation occurs at 48h for both BL cell lines (S5D, S5E Fig). These observations across a time course of treatment revealed a continuum in possible responses ranging from reactivation to cell death. These results also highlight the heterogeneity of responses to distinct HDAC inhibitors and various concentrations across malignancy models.

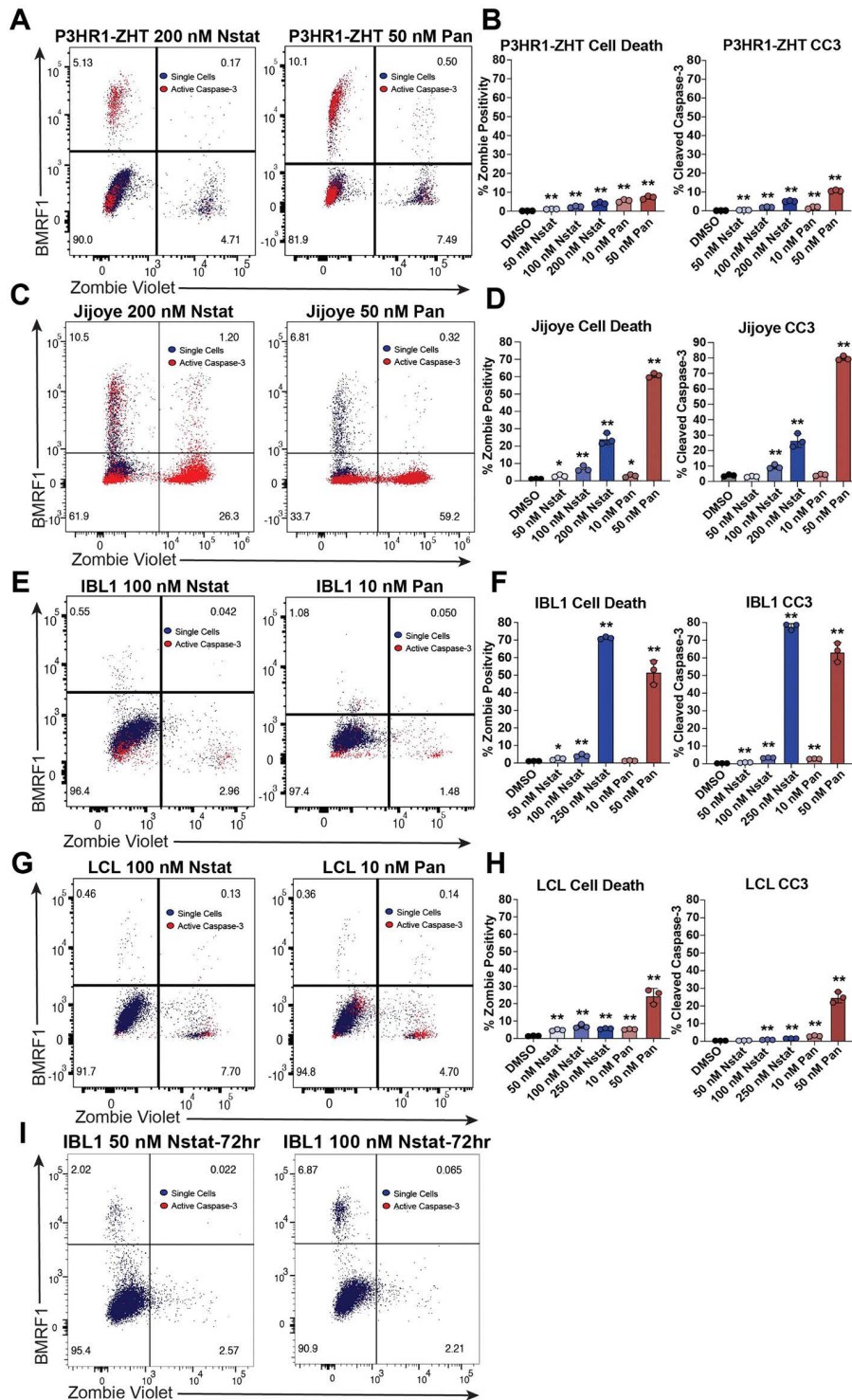

**Fig 3. HDAC inhibition leads to distinct fate outcomes in EBV+B cells.** (A) P3HR1-ZHT cells were treated with panobinostat and nanatinostat for 48h and subsequently stained for Zombie Violet, cleaved caspase-3, and BMRF1 to assess fate outcome of HDAC inhibitor treated cells. Blue cells are single cells, and red cells are cleaved caspase-3 positive. (B) Percentage of dead P3HR1-ZHT cells and cleaved caspase-3 positive P3HR1-ZHT cells following treatment with varying doses of nanatinostat and panobinostat. (C) Same experimental setup as panel A with Jijoye cells. (D) Percentage of dead and cleaved caspase-3 positive Jijoye cells following treatment with varying doses of nanatinostat and panobinostat. (E) Same experimental setup

as panel A with IBL1 cells. (F) Percentage of dead and cleaved caspase-3 positive IBL1 cells following treatment with varying doses of nanatinostat and panobinostat. (G) Same experimental setup as panel A with an LCL. Percentage of dead and cleaved caspase-3 positive LCLs following treatment with various doses of nanatinostat and panobinostat. (I) IBL1 cells were treated with increasing doses of nanatinostat for 72h and stained for Zombie Violet, cleaved caspase-3, and BMRF1 to assess fate outcome and lytic reactivation following longer treatment time. For panels B,D,F,H, $n = 3$ *$p < 0.05$, **$p < 0.01$.

## HDAC inhibitors lead to expression of the immediate-early BZLF1 gene in a population of cells, but incomplete expression of the full lytic cycle

To explore the heterogeneity and temporal dynamics of lytic reactivation, we used the BL line Jijoye and RNA Flow FISH to simultaneously probe for expression of an IE gene (*BZLF1*), an early gene (*BGLF4*), and a late gene (*BLLF1*) following HDAC inhibition. This technique allows for single-cell resolution of expression of multiple genes and is extremely targeted for each gene of interest. Treatment with either nanatinostat or panobinostat for 48h led to expression of BZLF1 in a comparable percentage of cells (Fig 4A, 4B). However, we observed a drop-off in the percentage of cells that went on to express an early or late gene following treatment with either HDAC inhibitor (Fig 4B). When we visualized the expression of both BZLF1 and BLLF1, a majority of cells that responded to HDAC inhibitor treatment only expressed the IE gene (Fig 4C, 4D). As expected, we observed that if a cell expressed a late gene (BLLF1), it will also express the IE gene BZLF1 (Fig 4C, 4D). These data suggest that HDAC inhibitors are effective at initiating the lytic cycle, but progression to the late stage of the lytic cycle is blocked in a portion of these cells. In contrast to HDAC inhibition, stimulating the inducible line P3HR1-ZHT with 4HT leads to a majority of cells progressing through the lytic cycle (BZLF1+, BLLF1+) compared to cells that only express an IE gene (S6 Fig). The threshold for successful lytic reactivation is inherently lower in this inducible model. These combined data highlight the importance of studying the dynamics of lytic reactivation with a biologically relevant stimulus.

## Utilizing single cell RNA sequencing to address underlying heterogeneity in lytic reactivation following HDAC inhibition

To further elucidate both host and viral gene expression heterogeneity following treatment with an HDAC inhibitor, we performed single cell RNA sequencing (scRNA-seq) on the P3HR1-ZHT line. We chose this line as it was responsive to HDAC inhibitors, and these data would complement our recently published work on lytic reactivation dynamics with the inducible stimulus 4HT [43]. We collected untreated P3HR1-ZHT cells and cells treated with 50 nM panobinostat for 48h, since this is when we observe peak lytic reactivation. We performed single-cell library prep and sequenced and aligned our data to both the human and EBV genome (Fig 5A). UMAP projections of samples based on treatment conditions showed distinct and significant changes in gene expression profiles (Fig 5B). When we visualized lytic gene expression based on the different stages of the lytic cycle, there was a clear progression through the lytic cycle. We observed that there were cells that only expressed IE or E lytic genes, and a smaller percentage of cells went on to express late lytic genes (Fig 5D). We also observed some cells in the untreated population that expressed genes associated with each lytic module, most likely corresponding to spontaneous lytic reactivation (Fig 5C).

## Applying scRNA-seq to identify host restriction factors for lytic reactivation

Cells were hierarchically clustered based on differential gene expression, and eleven unique clusters were identified (Fig 6A). Cluster 8 was our identified lytic population while clusters 0, 3, and 10 did not reactivate the lytic cycle following HDAC inhibitor treatment. Spontaneous lytic cells were associated with cluster 9, while the rest of the clusters were from majority unstimulated cells. When we calculated the number of unique genes expressed and total RNA counts in each cluster, lower RNA counts in cluster 8 signified host shutoff during lytic reactivation (S7A Fig) [43]. We identified cluster 3 as our abortive lytic cluster based on host gene expression patterns and lack of late lytic gene expression. Similar to our

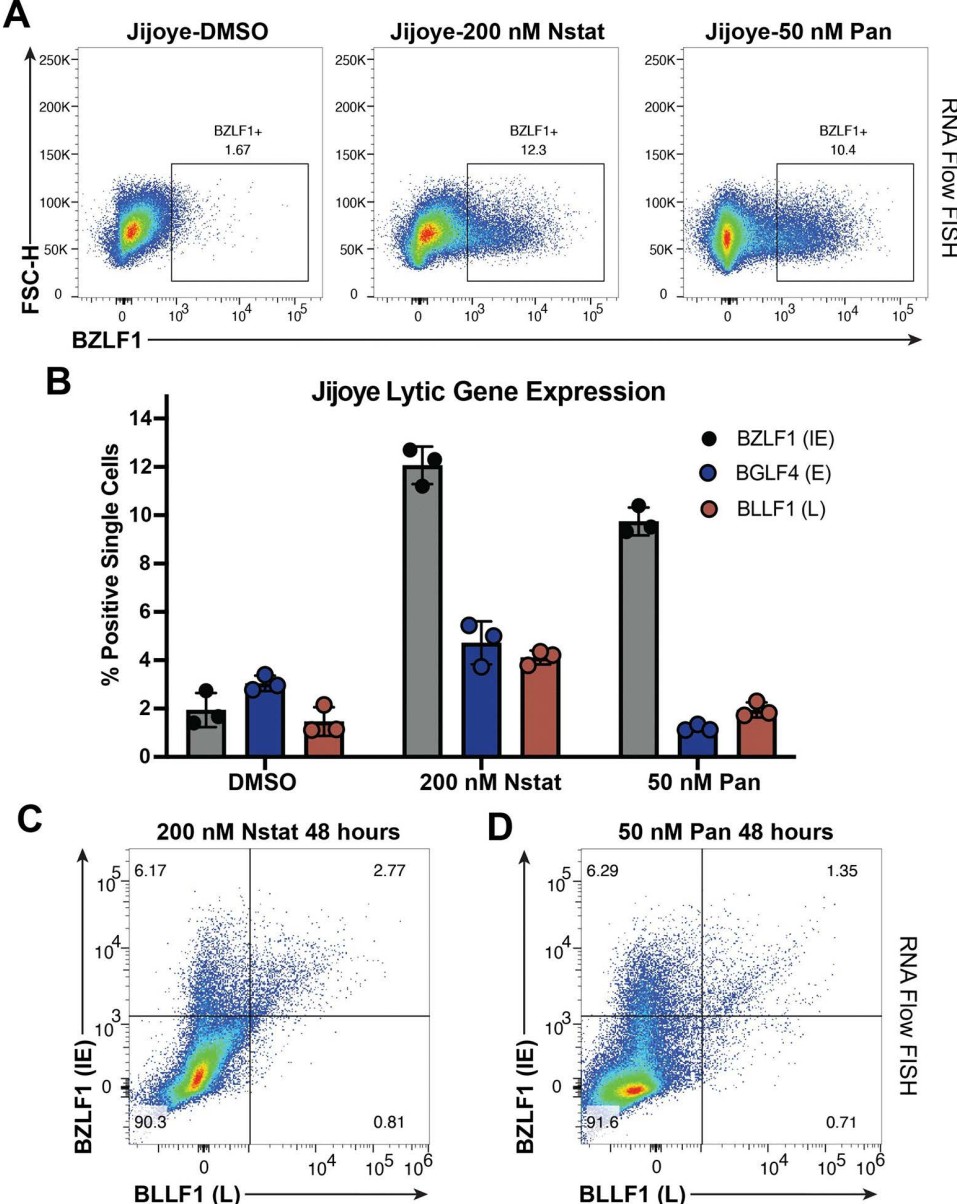

**Fig 4. Treatment with an HDAC inhibitor leads to incomplete expression of the entire EBV lytic cascade.** (A) RNA Flow FISH plot representing expression of the EBV lytic gene BZLF1 in Jijoye cells treated with DMSO, 200 nM nanatinostat or 50 nM panobinostat for 48h. (B) Percentage of single Jijoye cells in a population that express an IE, E or L gene following treatment with DMSO, 200 nM nanatinostat, or 50 nM panobinostat for 48h. (C) RNA Flow FISH plot of cells that express the IE gene BZLF1 and the L gene BLLF1 following treatment with 200 nM nanatinostat for 48h. (D) Same experimental procedure except for treatment with 50 nM panobinostat.

previous findings, high NF-κB activity was specific to this cluster as well as targets of NF-κB signaling such as *VCAM1* and *BCL2* (Figs 6B, S7B) [43]. Additionally, this phenotype is independent of LMP-1 expression which is known to drive expression of NF-κB (S7B Fig) [47,48]. Expression of other EBV latency genes were also enriched in the lytic cluster, including the EBNA3s and LMP2A (S7B, S7C Fig). LMP2A is known to restrict lytic reactivation via BCR crosslinking, but we do not see expression of this gene in the abortive cluster, indicating host factors are important for this phenotype. To

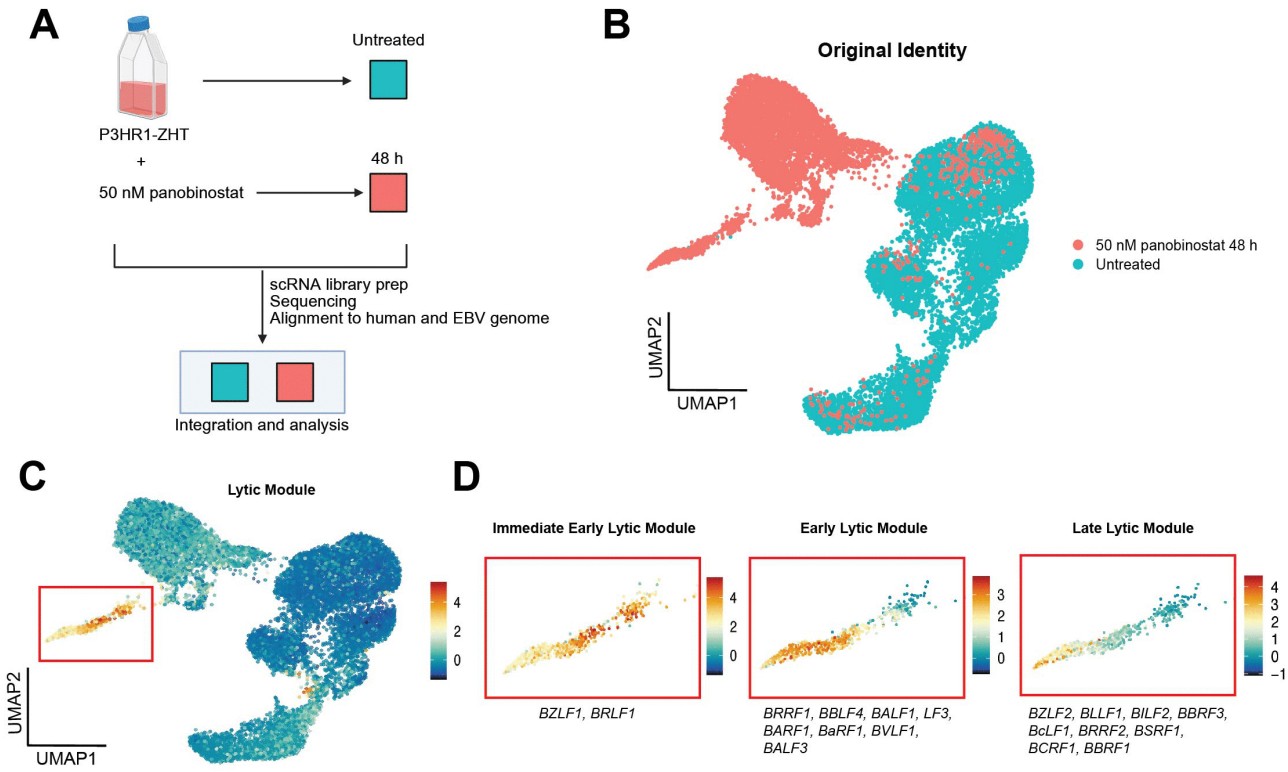

**Fig 5. Single-cell RNA sequencing reveals unique populations of lytic reactivation following treatment with HDAC inhibitor.** (A) Experimental setup for sc-RNA seq experiment. P3HR1-ZHT cells were treated with 50 nM panobinostat or untreated for 48h. Cells were harvested, and single cell libraries were prepared. Libraries were sequenced and reads were aligned to a concatenated human and type 2 EBV genome. Treatment timepoints were integrated for downstream analysis (made with Biorender). (B) Following pre-processing and quality control filtering, PCA analysis was run, and cells were projected on a UMAP based on their original identity of treated or untreated. Overall, most treated cells clustered independently of untreated cells. (C) Lytic viral gene expression module encompassing expression of genes from IE, E, and Late stages of the lytic cycle. The lytic cluster is denoted by a red box, and all included genes are listed in D. (D) Viral gene expression modules separated by stage of lytic cycle. Immediate early (IE) gene expression module: *BZLF1*, *BRLF1*. Early gene expression module: *BRRF1*, *BBLF4*, *BALF1*, *LF3*, *BARF1*, *BaRF1*, *BVLF1*, and *BALF3*. Late gene expression module: *BZLF2*, *BLLF1*, *BILF2*, *BBRF3*, *BcLF1*, *BRRF2*, *BSRF1*, *BCRF1*, and *BBRF1*. Created in BioRender. Barry, A. (2026) https://BioRender.com/x6wllot.

this end, we observed expression of host genes involved in immune signaling pathways including *CD137*, *STAT1*, and *CCR7* enriched in the abortive lytic cluster (Fig 6B). Our model for individual cell fates following reactivation aligns with our previous experiments wherein a small percentage of cells will reactivate the lytic cycle, but many cells will be refractory to treatment or exhibit an abortive lytic phenotype (Fig 6C). Both the abortive (cluster 3) and the refractory (cluster 0) population express similar genes, however expression of immune signaling genes and NF-κB targets are much higher in the abortive lytic cluster (cluster 3 vs cluster 0) (Fig 6B).

To compare our dataset with previous research, we examined the expression pattern of both *MYC* and *STAT3* as these have been shown to be host restriction factors for EBV lytic reactivation [49–51]. We observed decreased expression of these genes in lytic cells and more uniform expression across untreated cells (Fig 6B). To validate these findings following HDAC inhibition, we performed RNA Flow FISH on P3HR1-ZHT cells treated with 50 nM panobinostat for 48h. Cells were probed for the expression of either STAT1 or STAT3 and the early lytic gene BGLF4. We chose to examine STAT1 since expression was high in the refractory and abortive lytic population (Fig 6B, 6E). We observed limited expression of either STAT1 or STAT3 in resting cells, but a significant increase in the percentage of cells that express both genes

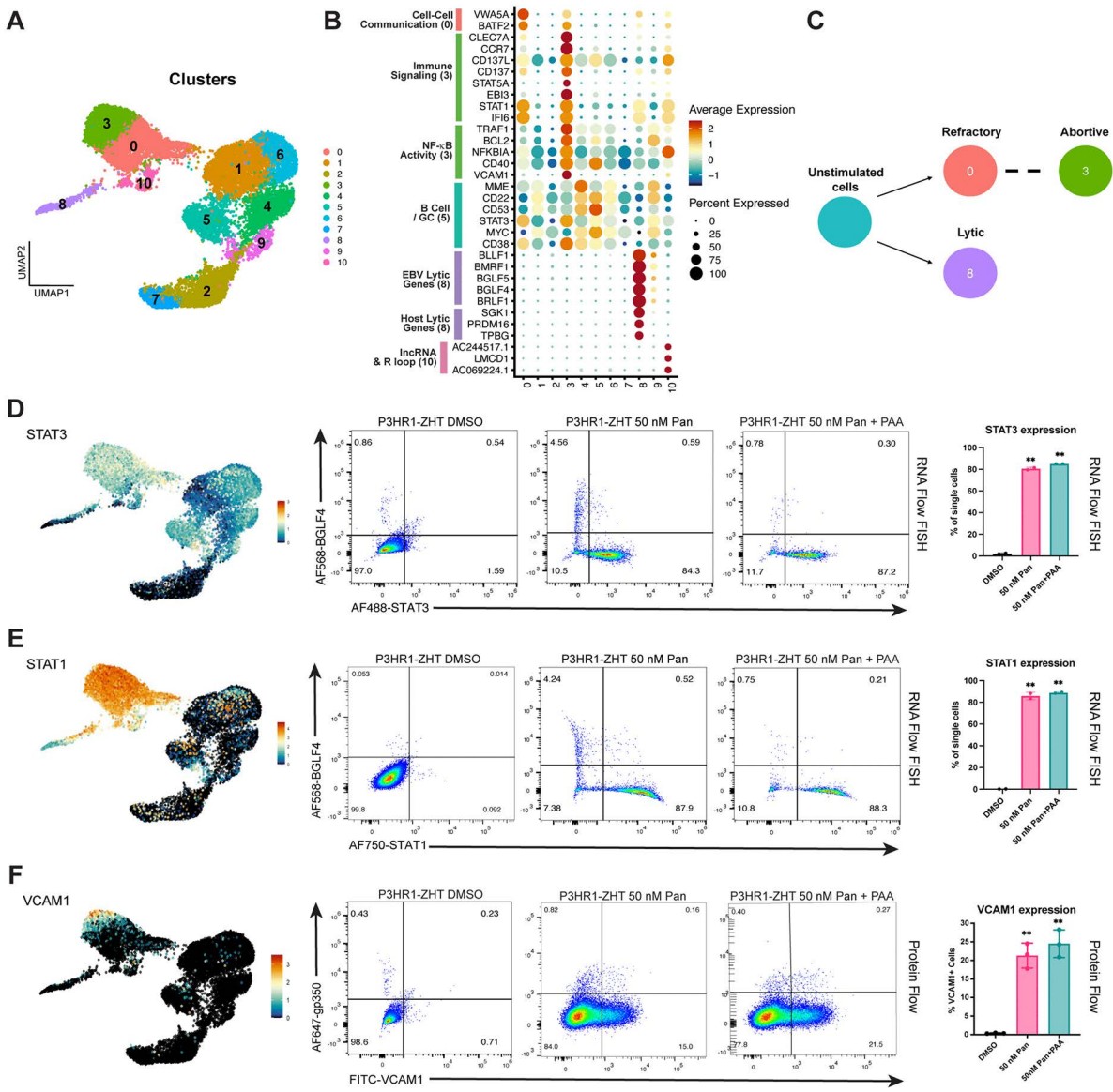

**Fig 6. scRNA-seq identifies potential host restriction factors for lytic reactivation in an HDAC inhibition background.** (***A***) UMAP of treated and untreated cells clustered via an unsupervised method. (***B***) Differential gene expression by cluster. Genes are annotated by their role and function determined by primary literature review and gene ontology analysis. (***C***) Model of cell trajectory following reactivation. Respective identities are denoted by their matching Seurat cluster. (***D***) UMAP expression of *STAT3* (left) and corresponding validation with RNA Flow FISH for expression of STAT3 and the early lytic gene BGLF4 in resting and panobinostat treated cells (middle). Replicates are represented by a bar graph which also shows STAT3 expression following treatment with PAA (right). (***E***) UMAP expression of *STAT1* (left) and corresponding validation with RNA Flow FISH for expression of STAT1 and the early lytic gene BGLF4 in resting and panobinostat treated cells (middle). Replicates are represented by bar graph which also shows STAT1 expression following treatment with PAA (right). (***F***) UMAP expression of *VCAM1* (left) and corresponding validation with protein flow cytometry for VCAM1 and the late lytic glycoprotein gp350 in resting cells and cells treated with panobinostat +/- PAA (middle). Replicates are represented by bar graphs (right). For panels D, E, n = 2 **p < 0.01 for panel F, n = 3 **p < 0.01.

following treatment with HDAC inhibitor (Fig 6D, 6E). As expected, expression of either host gene was anti-correlated with cells that expressed the early lytic gene BGLF4. When we inhibited viral DNA replication with PAA, we did not observe a change in the percentage of cells that expressed either STAT1 or STAT3 (Fig 6D, 6E). This phenotype likely indicates that

any potential signaling occurs before viral DNA replication. As expected, MYC expression was also anti-correlated with successful lytic reactivation (S7D Fig). Interestingly, expression of STAT1 as a marker for abortive lytic cells was unique to HDAC inhibitor treatment, whereas STAT3 was expressed in abortive cells following treatment with either 4HT or an HDAC inhibitor (S7E Fig).

We were also interested in validating the gene *VCAM1* as expression was highly specific to our identified abortive cluster (Fig 6B, 6F). VCAM1 is a vascular cell adhesion molecule that can be expressed on B cells following NF-κB activation [52]. To validate our scRNA-seq data, we performed extracellular protein staining for VCAM1 and the late lytic glycoprotein gp350 following reactivation. We observed no expression of VCAM1 in resting cells but observed a significant increase in the expression of this protein following treatment with 50 nM panobinostat (Fig 6F). Additionally, the expression of gp350 and VCAM1 were mutually exclusive, which makes VCAM1 a reliable marker for identifying abortive cells. There was a slight, but insignificant, increase in the percentage of cells that express VCAM1 following the addition of PAA (Fig 6F). Similar to our finding with STAT1 expression, expression of VCAM1 in abortive lytic cells was specific to HDAC inhibitor treatment. We observed no VCAM1 expression in P3HR1-ZHT cells stimulated with 4HT (S7F Fig).

## Identification of signaling pathways that are associated with unsuccessful lytic cycle progression

Our previous and current data supports that NF-κB activity is upregulated in abortive lytic cells. In the context of HDAC inhibition, VCAM1 is a reliable marker, while we utilized ICAM1 as a proxy for NF-κB activity and a marker of abortive lytic cells following 4HT treatment. When we previously treated cells with an IKK inhibitor, a key component of NF-κB, we did not observe a change in the percentage of cells that went lytic, even though NF-κB activity was abrogated [43]. Based on this data and our current findings regarding the expression of VCAM1 and immune signaling genes like *STAT1* and *STAT3*, we were interested in identifying potential signaling pathways that might be acting upstream of these factors. To this end, we used a tool called CellChat that uses scRNA-seq data and published human signaling interactions to compute communication probability and infer cellular communication networks [46]. We were particularly interested in signals originating from cluster 3 as this is where abortive host markers were most strongly expressed. CellChat analysis identified several predicted signaling pathways that originated from the abortive cluster 3 (S8A Fig). We focused on the CD137-CD137L pathway as this signal was predicted to originate from cluster 3 and signal to both itself as well as the refractory cluster 0 (Fig 7A). Additionally, there was some predicted signaling to the lytic cluster (cluster 8). CellChat predicted that the strongest signaling happened within cluster 3, with some potential cells in cluster 0 receiving or influencing the signal (Fig 7D).

CD137, also known as TNFRSF9 or 4–1BB, and its ligand CD137L are members of the tumor necrosis factor (TNF) family. Typically, the ligand is expressed on the surface of antigen presenting cells while the receptor is typically expressed on T cells [53,54]. CD137/CD137L interaction can lead to the activation of various downstream signaling pathways including NF-κB, which leads to pro-survival signaling in the T cell [55–57]. It has also been observed that various types of B cell malignancies can express CD137L, including Burkitt lymphomas [58]. There has also been evidence that EBV can drive expression of CD137 in both NK/T cell lymphomas and Hodgkin lymphomas through LMP-1 [59,60]. Expression of CD137 in this context has been linked to both the pathogenesis and persistence of cancer cells. We therefore hypothesized that CD137 signaling in abortive lytic cells plays a role in preventing successful progression through the lytic cycle.

When we visualized gene expression of *CD137*, the receptor was expressed heterogeneously across treated cells (clusters 0 and 3) (Fig 7B). The ligand *CD137L* had more overall expression in both treated and untreated cells. However, some of the strongest expression of *CD137L* came from abortive lytic cells in cluster 3 (Fig 7C). To validate this predicted signaling at the protein level, we stained for the presence of either CD137 or CD137L as well as the late lytic glycoprotein gp350. We observed no CD137 expression in resting cells but a significant increase in expression following treatment with panobinostat (Fig 7B). Additionally, expression of CD137 was anticorrelated with gp350 expression. When we stained for CD137L, we observed some expression basally, but this increased when cells were treated with panobinostat (Fig 7C).

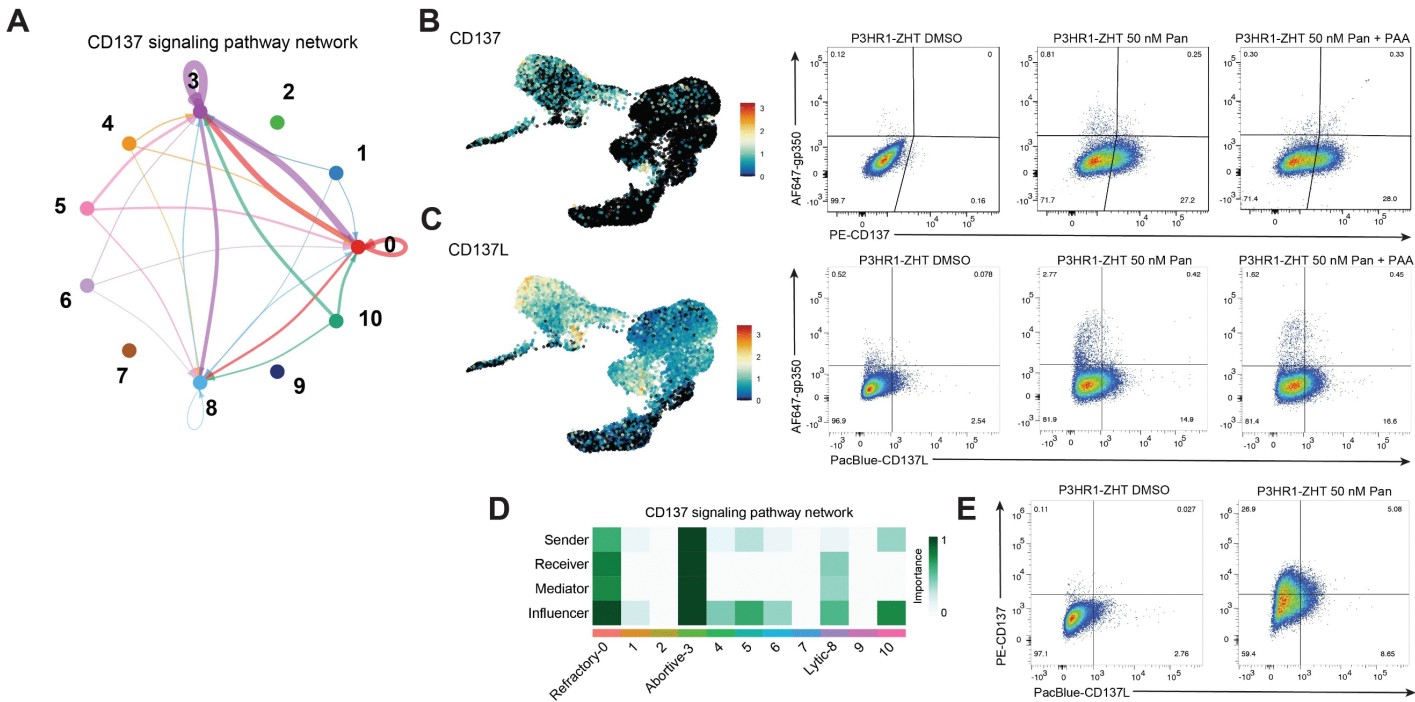

**Fig 7. scRNA-seq and CellChat analysis reveal CD137 signaling as a potential pathway driving the abortive lytic phenotype.** (*A*) Hierarchy plot of the CD137 signaling pathway that depicts predicted signaling to and from each cluster. Loops denote signaling within the same cluster. Thicker bands correspond to stronger predicted overall signaling. (*B*) UMAP expression of *CD137* (left) and corresponding validation by protein flow cytometry for CD137 and gp350 in resting cells and cells treated with panobinostat +/- PAA (right). (*C*) UMAP expression of *CD137L* (left) and corresponding validation by protein flow cytometry for CD137L and gp350 in resting cells and cells treated with panobinostat +/- PAA (right). (*D*) Network centrality scores were computed to visualize which clusters were dominant senders (sources) and receivers (targets) in the CD137 signaling pathway. (*E*) Flow cytometry plots of resting cells and cells treated with panobinostat for expression of both CD137 and CD137L.

Expression of the ligand was also anticorrelated with lytic reactivation. Addition of PAA did not lead to a significant change in the percentage of cells that expressed either the receptor or the ligand (Fig 7B, 7C). When we stained for both the receptor and the ligand simultaneously, we observed that a majority of the population expressed either the receptor or the ligand (Fig 7E). However, there was a small proportion of cells that expressed both the receptor and the ligand.

To ascertain whether this phenotype was specific to HDAC inhibition, we stimulated P3HR1-ZHT cells with 4HT and stained for CD137, CD137L, and gp350. We observed a similar staining pattern where lytic reactivation was mutually exclusive from either CD137 or CD137L expression. However, there was not a population of cells that expressed both the receptor and the ligand (S8B Fig). Overall, the percentage of cells that expressed either CD137 or CD137L was lower following 4HT treatment compared to HDAC inhibition. These data show that HDAC inhibition strongly upregulates members of the CD137/CD137L signaling pathway. Additionally, with most HDAC inhibitor treated cells expressing either the receptor or the ligand, this suggests that signaling is occurring between different abortive lytic cells.

## CD137 signaling is involved in preventing successful lytic reactivation

To determine whether CD137 signaling played an active role in preventing successful lytic reactivation, we utilized a Cas9-RNP approach to abrogate expression of CD137, CD137L, or both the receptor and ligand simultaneously. We transfected P3HR1-ZHT cells with Cas9 and sgRNAs targeting our genes of interest in addition to an sgRNA targeting the host gene CD46 (Fig 8A). CD46 is a non-essential surface protein expressed on the surface of B cells and is used as a proxy for

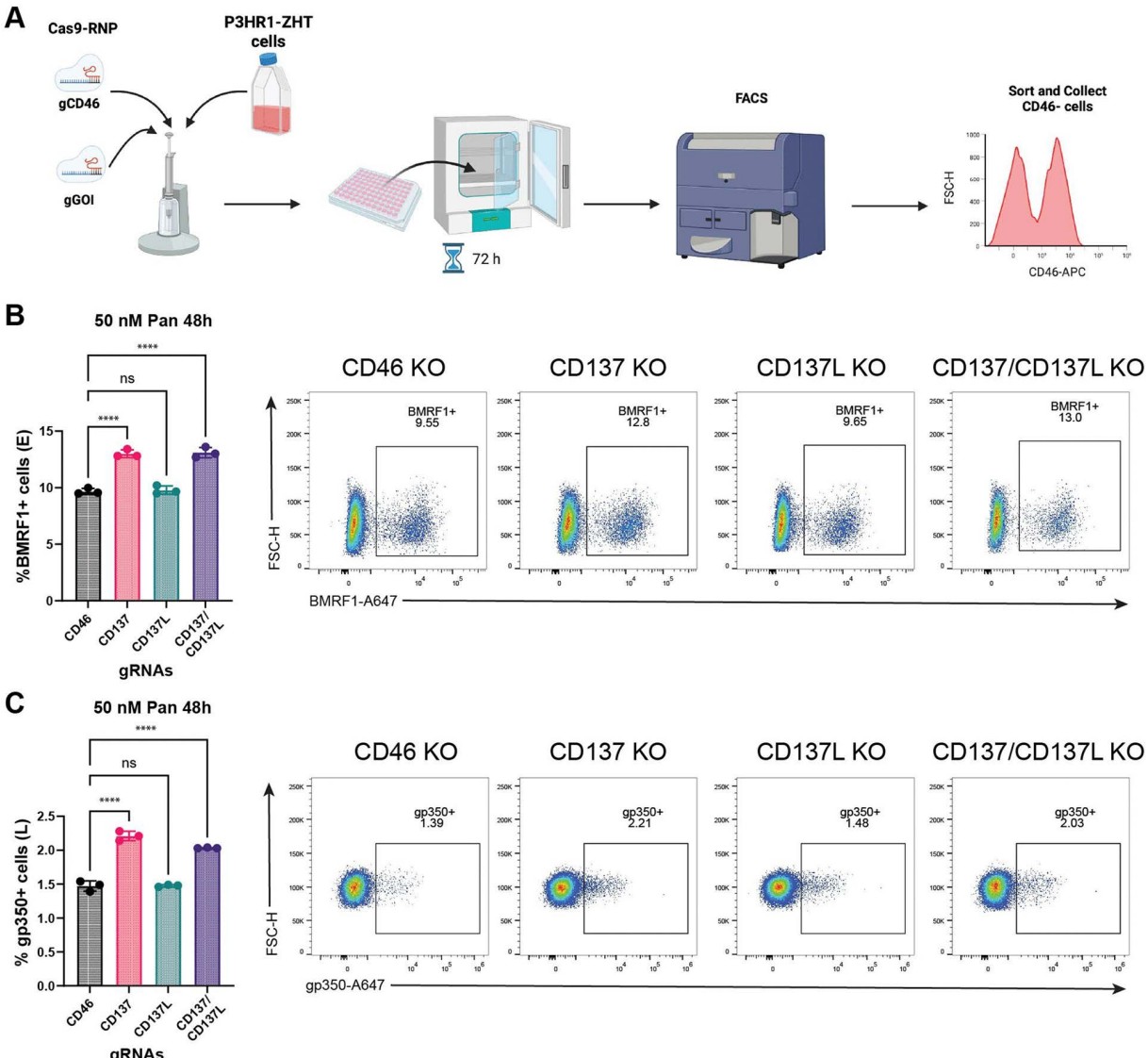

**Fig 8. Knockout of CD137 leads to an increase in lytic protein expression at both the early and late stage of reactivation.** (A) Workflow of knock-out approach. P3HR1-ZHT cells were transfected with Cas9 and sgRNAs targeting CD46 and CD137, CD137L, or CD137 + CD137L. CD46 is a non-essential surface protein that is used as a proxy for successful knockout. Cells recover for 72h following transfection. Cells are then stained for CD46 and sorted on the CD46-negative population, enriching for knockout cells (made with Biorender). (B) P3HR1-ZHT knockout cells were stimulated with 50 nM panobinostat for 48h and stained for the early lytic protein BMRF1. One-way ANOVA was performed comparing values to the CD46 only knockout. (C) P3HR1-ZHT knockout cells were stimulated with 50 nM panobinostat for 48h and stained for the late lytic protein gp350. One-way ANOVA was performed comparing values to the CD46 only knockout. For panels B, C, n = 3 **** p < 0.0001. Created in BioRender. Barry, A. (2026) https://BioRender.com/x0mj6fx.

successful transfection for our guides of interest [61]. Cells were stained for CD46, and CD46-negative cells were sorted and collected, enriching for a successfully knocked out population. Downstream protein flow cytometry and sequencing were performed to validate successful knockout for our guides of interest (S9 Fig). Knockout cells were treated with 50 nM panobinostat for 48h and stained for the early lytic protein BMRF1. In the CD46-only knockout (non-targeting control) ~9.5% of cells expressed BMRF1. In the CD137 knockout, the percentage of cells that expressed BMRF1 increased to

~13%. Interestingly, in the CD137L knockout, the percentage of BMRF1+cells was roughly equivalent to the non-targeting control (CD46 KO). The double knockout of the receptor and the ligand phenocopied the receptor only knockout (Fig 8B). We also examined the expression of the late lytic protein gp350 to determine if a higher percentage of cells progressed through the lytic cycle. In the CD46-only knockout, ~1.4% of cells expressed gp350. In the CD137 knockout, the percentage of gp350+cells increased to ~2.2% (Fig 8C). The various knockouts behaved similarly between stages of the lytic cycle. The CD137L knockout had no effect, and the double knockout had a similar percentage of lytic cells as the receptor only knockout (Fig 8C).

Overall, the CD137 receptor knockout led to a 9.55% to 12.8% change (~70% increase) in the percentage of early lytic cells and a 1.39% to 2.21% change (~60% increase) in the percentage of late lytic cells (Fig 8B, 8C). While we observed a significant increase in lytic cells that progressed through the lytic cycle at both the early and late stage, we still saw a drop-off in cells that progressed from the early to the late stage of reactivation. Additionally, late lytic cells accounted for ~2.5% of the total fully lytic population. While CD137 receptor signaling does have an observable effect on lytic reactivation at multiple stages, other host factors and signaling pathways likely play a redundant or additive role in the absence of CD137 signaling.

## Discussion

Lytic reactivation therapies are an attractive treatment option for EBV-associated cancers as they can be more targeted and less toxic than traditional cancer treatments. HDAC inhibitors are one example of a pharmacological agent that can be used to reactivate EBV. While previous work has shown that combination therapy leads to greater cell/tumor death, the dynamics of lytic reactivation responsible for this response has been less well explored [8,10,13,14]. Through multiple approaches at the RNA and protein level we were able to better characterize lytic reactivation following HDAC inhibition. On a larger scale, we observed heterogeneous responses between EBV positive cell lines that represent distinct cancer backgrounds and heterogeneous responses within each cell line. The data described here provide a snapshot of what occurs within a population of EBV infected cells responding to HDAC inhibitors. Applying single cell analysis rather than bulk measurements provides a lens into the heterogeneous response within a population. However, the EBV lytic cycle is an extremely dynamic process and a single snapshot into the cell cycle or lytic gene expression is not able to completely capture the complexity of the process.

The BL line Jijoye and its subclone line P3HR1 were the most responsive cell lines to HDAC inhibitors and exhibited the highest overall reactivation rate. Additionally, both lines responded in a dose dependent manner with increased concentrations leading to a higher percentage of cells expressing lytic proteins. This lytic reactivation was observed regardless of high levels of cell cycle arrest, cell death, or cleaved caspase-3 (Figs 2, 3). Across experiments, these lines exhibited peak reactivation at 48h following treatment with HDAC inhibitors. It is important to note that traditional human BL tumors and cell lines typically exhibit a latency I background, where EBNA1 is the only EBV latency protein expressed. P3HR1 is EBNA2 deleted and exhibits Wp-restricted latency but has previously been used in other work to study HDAC inhibitor induced EBV lytic reactivation [62,63]. Jijoye displays a type III latency pattern and expresses all the EBV latency proteins, similar to an LCL. While these cell lines may not be representative of all BLs, our lab has recently shown that newly derived endemic Burkitt lymphoma lines display a variety of latency patterns, including expression of EBNA2 and EBNA3A [64]. Our data underscore the importance of understanding lytic reactivation in various backgrounds as they respond uniquely to HDAC inhibition regardless of latency type.

The AIDS immunoblastic lymphoma cell line (IBL1) and the LCL were overall less responsive to both HDAC inhibitors and were more sensitive to cell cycle arrest and cell death, particularly following treatment with panobinostat. However, IBL1 was more responsive to nanatinostat when treated for a longer time interval. This heterogeneity across cell lines was seen at multiple levels including differences in cell cycle arrest, lytic protein expression, cell fate following reactivation, and the time it took for cells to reach 'peak' reactivation.

The concentrations of HDAC inhibitor and the time course we used in this study are useful for studying cellular dynamics following treatment. However, these treatments *in vitro* are different than those used *in vivo*. In the most recent phase 2 clinical trial of nanatinostat, the treatment regimen included 20 mg nanatinostat once a day for four days a week. Valganciclovir was administered daily and the entire regimen followed a 28-day cycle [41]. At this concentration of nanatinostat, the observed Cmax from patient plasma was 186 ng/mL or approximately 471 nM [41,65]. The concentrations we used in these experiments reflect a careful balancing act between cell cycle arrest, cell death, and potential lytic reactivation. In the P3HR1-ZHT and Jijoye line, a single dose of nanatinostat led to accumulating levels of cell death and active caspase-3 over time (S5D, S5E Fig).

Our early experiments with HDAC inhibitors showed that all treated cell lines arrested at some level, with a decrease in the number of cells in S phase and an increase in the percentage of cells in either G1 or sub-G1 populations. This cell cycle arrest is part of the DNA damage response (DDR) and the tumor suppressor p53 is known to be required for the halt in G1 [66]. Previous research into the mechanism of action of HDAC inhibitor induced viral reactivation found that p53 is indispensable, highlighting the importance of the p53-dependent DDR in reactivation [67]. EBV lytic reactivation also leads to a DDR likely initiated by viral DNA structures as well as BGLF4 phosphorylation of Tip60, leading to activation of ATM and p53 [68,69]. While this DDR occurs, a pseudo-S phase is created allowing for viral DNA replication [28,70,71]. When we stained HDAC inhibitor treated P3HR1-ZHT and Jijoye cells for BrdU, 7AAD, and the early lytic protein BMRF1, we indeed saw lytic cells in G1/S phase, denoting active viral DNA replication in this pseudo-S phase. It is apparent that there is a fine balance between HDAC inhibitor induced DNA damage and EBV lytic reactivation associated DNA damage signaling in creating a suitable environment for reactivation. For example, we observed a range of lytic reactivation in each cell line between experiments (**Figs 2**, **3**). This is most likely explained by the different staining protocols required of each experiment but could also be due to biological variation and the timing of the cell cycle within a population of cells at the time of treatment.

Staining treated cells simultaneously for BMRF1, cleaved caspase-3, and a cell death marker (Zombie Violet) revealed a cell fate continuum. Within a population of cells there were distinct fates spanning reactivation, cell death, or a refractory phenotype. It is not clear what may be driving individual decisions between these outcomes. Interestingly, of the cell lines to respond best to HDAC inhibitor treatment, we saw that a greater percentage of lytic (BMRF1+) P3HR1-ZHT and Jijoye cells were also positive for cleaved caspase-3 following treatment with both HDAC inhibitors (**Fig 3A**, **3C**). It has been shown that EBV can use active caspases, including caspases -3, -6, and -8, in non-canonical ways to assist in viral reactivation [72]. More recent studies have also observed that knockdown of TXNIP- a component upstream of inflammasome mediated cleavage of caspase-1- or knockdown of caspase-1 directly reduces EBV lytic reactivation [73,74]. It is possible that the BL cell lines are more responsive to HDAC inhibitor treatment and undergo lytic reactivation at a higher percentage than IBL1 cells or LCLs due to the utilization of this enzyme.

While the BL cell lines were most responsive overall, only a fraction of cells within a population went fully lytic and expressed a late lytic gene. From our RNA Flow FISH experiments, we observed that HDAC inhibitors were successful at activating expression of *BZLF1*, but that this did not guarantee downstream activation of a late lytic gene. Of the currently published findings, much of the work has focused on identifying how HDAC inhibitors activate immediate-early genes. While it was found that p53 was important for reactivation, it was also observed that in a p53 knockdown setting overexpression of Zta or Rta was sufficient to reactivate EBV [67]. Additionally, the finding that PKCδ is important for HDAC inhibitor mediated reactivation also revealed that Sp1 can bind to the Z promoter and interact with the protein kinase [37]. With these findings in mind, this halt in progression of the lytic cycle that we observe must occur before viral DNA replication. This is also corroborated by our findings that expression of potential host restriction factors like STAT1, STAT3, and VCAM1 does not change when viral DNA replication is inhibited with PAA (Fig 6D-6F).

Viral DNA replication is dependent on the function of early lytic genes and must successfully transpire for expression of late lytic genes. Our results suggest that hyperacetylation is effective at activating BZLF1, but it does not guarantee a

complete lytic reactivation cascade. It is possible that Zta activity itself can lead to a phenotype that prioritizes cell growth and resistance to apoptosis over progression through the lytic cycle, especially in the context of cancer. Zta can directly bind to sequences in the human IL-10 promoter, which promotes growth and enhanced viability of B cells [75]. Zta can also directly downregulate the expression of tumor necrosis factor receptor 1 (TNFRSF1A), which prevents downstream TNFα induced apoptosis [76,77]. This abortive lytic phenotype has been observed by our group and others, and there have been investigations into whether host factors may be involved in keeping EBV-positive cancer cells refractory to lytic reactivation [43,49,78,79].

Performing scRNA-seq on cells treated with HDAC inhibitor allowed us to better address this question. Similar to our previously published findings, we observed high NF-κB activity in abortive lytic cells. It has been established that the EBV latent protein LMP1 can directly lead to downstream activation of NF-κB and that this activity can inhibit lytic reactivation [47,80]. However, in our single cell dataset, we see LMP1-independent expression of NF-κB (S7B Fig). This suggests that other factors may be involved in upregulating NF-κB family members following treatment with reactivation stimuli. We were interested in what may be driving this phenotype since we also previously observed that inhibiting NF-κB activity directly with an IKK inhibitor did not affect overall lytic reactivation [43].

There are multiple host signaling pathways that can lead to the activation of NF-κB, many involved in the identification of pathogens. Indeed, both RNA-sensing and DNA-sensing pathways have been implicated in the identification of EBV RNAs/proteins [81–84]. Additionally, NF-κB activity itself can lead to diverse phenotypes including pro-survival signaling and anti- or pro-inflammatory signaling depending on the context [85]. This is reflected in our results that abortive lytic cells express factors like STAT3 and VCAM1. Recent studies have also shown a link between lytic reactivation and the NLRP3 inflammasome. A balance has been observed where the inflammasome is activated in early stages of lytic reactivation while late lytic proteins have shown to be involved in inhibiting the inflammasome [74,86,87]. Within a population of cells there is a balance between immune signaling/sensing of the virus and successful lytic reactivation, and this is likely what drives heterogeneous cell fates. We therefore hypothesized that multiple immune signaling pathways were likely responsible for this abortive lytic population. By combining our scRNA-seq data with CellChat analysis, we were able to narrow down potential signaling pathways that drive this phenotype.

We focused on signaling that originated from our abortive lytic cells and identified the CD137/CD137L pathway. We observed that abortive lytic cells express either the ligand or the receptor, and that a smaller percentage of treated cells express both receptor and ligand (Fig 7). This phenotype was also stronger in HDAC inhibitor treated cells compared to P3HR1-ZHT cells treated with the inducible stimulus 4HT. This expression pattern provides evidence for receptor/ligand interaction between abortive lytic cells and the subsequent NF-κB activity we observe. This signaling activity in turn leads to an immune evasion and pro-survival phenotype rather than successful lytic reactivation. This aligns with the fact that in a latent state EBV expresses very few genes and is therefore successful at evading immune detection.

To test this hypothesis, we utilized a Cas9-RNP approach to knockout the expression of CD137 and CD137L. We observed that abrogating expression of the receptor CD137 led to a significant increase in the percentage of cells that expressed early (BMRF1) and late (gp350) lytic proteins following HDAC inhibitor treatment (Fig 8). Interestingly, targeting the expression of the ligand CD137L did not lead to a change in the percentage of lytic cells compared to the non-targeting control (Fig 8). It is possible that other ligands, like extracellular matrix proteins, could potentially be signaling through the CD137 receptor, and this is why we see no change in lytic reactivation in the absence of CD137L [88]. Additionally, there could be potential ligand-independent activation of the receptor following rapid upregulation post HDAC inhibitor treatment. There has been investigation into pre-ligand assembly domains (PLADs) within the TNF family of receptors and their role in receptor stabilization in a ligand-independent manner [89]. It is important to note that our data is captured at a moment in time when in reality there are many dynamic processes at play. These proteins are not constitutively expressed but upregulated following treatment with a lytic reactivation stimulus, and it is still unclear how timing plays a role in downstream signaling. Additionally, this *in vitro* work is inside a closed system without the interaction of

other immune cells. While we observe expression of both CD137 and CD137L on EBV-positive B cells, *in vivo* these surface proteins could potentially interact with T and NK cells under reactivation conditions.

Uncovering what is driving these heterogeneous responses during reactivation will have important repercussions for clinical applications and treatment regimen in the future. Understanding which cancer types respond best to this treatment can lead to better clinical outcomes as well. Additionally, this analysis and further investigation into what makes EBV refractory to complete reactivation can lead to improved effectiveness of the downstream antiviral treatment in lytic reactivation therapies. It is clear that multiple host factors and various signaling pathways play a role in preventing EBV lytic reactivation at different stages of the lytic cycle. There is more work to be done in understanding the complex and interconnected immune signaling that occurs during EBV lytic reactivation.

## Supporting information

**S1 Fig. Relative S phase reduction and sub-G1 cells following HDAC inhibition.** (*A*) Relative percentage of P3HR1-ZHT cells in S phase and after treatment with HDAC inhibitors for 24- and 48h. Values normalized to the percentage of cells in S phase after DMSO treatment. Percentage of P3HR1-ZHT cells in the sub-G1 population following treatment with HDAC inhibitors. (*B*) Relative percentage of Jijoye cells in S phase and after treatment with HDAC inhibitors for 24- and 48h. Values normalized to the percentage of cells in S phase after DMSO treatment. Percentage of Jijoye cells in the sub-G1 population following treatment with HDAC inhibitors. (*C*) Relative percentage of IBL1 cells in S phase and after treatment with HDAC inhibitors for 24- and 48h. Values normalized to the percentage of cells in S phase after DMSO treatment. Percentage of IBL1 cells in the sub-G1 population following treatment with HDAC inhibitors. (*D*) Relative percentage of an LCL in S phase and after treatment with HDAC inhibitors for 24- and 48h. Values normalized to the percentage of cells in S phase after DMSO treatment. Percentage of LCLs in the sub-G1 population following treatment with HDAC inhibitors. For all panels, error bars show data ±SD, n = 3. *p < 0.05, **p < 0.01.
(TIF)

**S2 Fig. Lytic flow plot replicates following HDAC inhibition.** (*A*) Gating strategy for BMRF1 + P3HR1-ZHT cells treated with increasing concentration of HDAC inhibitor for 48h. (*B*) Gating strategy for BMRF1 + Jijoye cells treated with increasing concentration of HDAC inhibitor for 48h. (*C*) Gating strategy for BMRF1 + IBL1 cells treated with increasing concentration of HDAC inhibitor for 48h. (*B*) Gating strategy for BMRF1 + LCLs treated with increasing concentration of HDAC inhibitor for 48h.
(TIF)

**S3 Fig. Cell cycle and lytic reactivation following 4HT treatment.** (*A*) Cell cycle plots with BrdU and BMRF1 staining of P3HR1-ZHT cells treated with 4HT for 24- and 48h.
(TIF)

**S4 Fig. Cleaved caspase-3 plot replicates following HDAC inhibition.** (*A*) Gating strategy for cleaved caspase-3 + P3HR1-ZHT cells treated with increasing concentration of HDAC inhibitor for 48h. (*B*) Gating strategy for cleaved caspase-3 + Jijoye cells treated with increasing concentration of HDAC inhibitor for 48h. (*C*) Gating strategy for cleaved caspase-3 + IBL1 cells treated with increasing concentration of HDAC inhibitor for 48h. (*B*) Gating strategy for cleaved caspase-3 + LCLs treated with increasing concentration of HDAC inhibitor for 48h.
(JPG)

**S5 Fig. Long time course HDAC inhibitor treatment.** (*A*) Flow plots of LCLs treated with high doses of nanatinostat for 72h and stained with the live/dead marker Zombie Violet, cleaved caspase-3, and the early lytic protein BMRF1. (*B*) LCLs were treated with high doses of nanatinostat for 72h and stained for the presence of BMRF1. (*C*) IBL1 cells were treated with increasing doses of nanatinostat for 72h and stained for the presence of the early lytic protein BMRF1. (*D*)

P3HR1-ZHT cells were treated with 200 nM nanatinostat for 72h. Cells were collected every 24h and subsequently stained for BMRF1, cleaved caspase-3, and cell death. (E) Jijoye cells were treated with 200 nM nanatinostat for 72h. Cells were collected every 24h and subsequently stained for BMRF1, cleaved caspase-3, and cell death.
(TIF)

**S6 Fig. Lytic RNA Flow FISH following 4HT treatment.** (A) P3HR1-ZHT cells were treated with DMSO or 100 nM HT for 24 h and collected for RNA Flow FISH analysis. Cells were probed for expression of the IE gene BZLF1 and the late gene BLLF1.
(TIF)

**S7 Fig. scRNA-seq QC and 4HT treatment comparison.** (A) QC RNA features by cluster. The number of unique RNAs expressed (genes, lncRNAs) per cell is depicted by nFeature_RNA. The total number of mapped reads per cell is depicted by nCount_RNA. (B) NFKB gene expression module: *NFKB1*, *NFKBIA*, *NFKBIZ*, *NFKB2*, *BCL2A1* and UMAP expression of the viral genes *LMP1* and *LMP2A*. Red circle denotes lytic cluster. (C) UMAP expression of the viral genes *EBNA3A*, *EBNA3B*, and *EBNA3C*. Red circle denotes lytic cluster. (D) UMAP expression of *MYC* and corresponding RNA Flow FISH validation probing for MYC and the late lytic gene BLLF1 following treatment with an HDAC inhibitor for 48h. (E) RNA Flow FISH plots of P3HR1-ZHT cells stimulated with 4HT for 24h and probed for expression of either STAT1 or STAT3 and the viral early lytic gene BGLF4. (F) Protein flow plot of P3HR1-ZHT cells treated with 4HT for 24h and stained for expression of VCAM1 and the late lytic protein gp350.
(TIF)

**S8 Fig. Potential signaling pathways identified by Cell Chat and 4HT treatment comparison.** (A) Predicted communication probability of the top signaling pathways originating from cluster 3 (abortive cluster) from CellChat analysis. CD137L-CD137 signaling pathway was the only hit to show potential signaling between all three clusters of interest: refractory (0), abortive (3), and lytic (8). (B) Protein flow plots of P3HR1-ZHT cells treated with 4HT and stained for CD137, CD137L, and gp350.
(TIF)

**S9 Fig. Validation of CD137, CD137L, and CD137 + CD137L knockouts.** (A) CD46 was knocked out in P3HR1-ZHT cells. Cells were stimulated with 50 nM panobinostat for 48h and stained for CD46, CD137, and CD137L. The CD46-positive and CD46-negative populations have comparable levels of CD137 and CD137L expressed. (B) CD46 and CD137 were knocked out in P3HR1-ZHT cells. Cells were stimulated with 50 nM panobinostat for 48h and stained for CD46, and CD137. The CD46-negative population expressed significantly reduced CD137 following treatment. (C) CD46 and CD137L were knocked out in P3HR1-ZHT cells. Cells were stimulated with 50 nM panobinostat for 48h and stained for CD46, and CD137L. The CD46-negative population expressed significantly reduced CD137L following treatment. (D) CD46, CD137, and CD137L were knocked out in P3HR1-ZHT cells. Cells were stimulated with 50 nM panobinostat for 48h and stained for CD46, CD137, and CD137L. The CD46-negative population expressed significantly reduced CD137 and CD137L following treatment. (E) All experimental KO lines were sorted for the CD46-negative population to enrich for successfully transfected cells. All lines were sequenced and compared to the CD46 only KO. The Synthego ICE score tool was used to provide a KO score.
(JPG)

**S1 Table. CC$_{30}$ and CC$_{70}$ values following HDAC inhibitor treatment.** CC$_{70}$ and CC$_{30}$ values for the cell lines P3HR1-ZHT, Jijoye, IBL1, and an LCL following treatment with panobinostat or nanatinostat for 48h.
(XLSX)

**S2 Table. sgRNAs used for CD137 and CD137L knockouts.**
(XLSX)

## Acknowledgments

We thank the Duke University School of Medicine for the use of the Flow Cytometry Shared Resource. Portions of figures were generated in Biorender. We also acknowledge Lyla Stanland for her help in training on the flow cytometers, Katie Willard for preparing the untreated P3HR1-ZHT library and training on the flow cytometers, and Elliott SoRelle for his assistance in computational training. We finally acknowledge the support of Viracta in providing nanatinostat. The content is solely the responsibility of the authors and does not necessarily represent the official views of the National Institutes of Health.

## Author contributions

**Conceptualization:** Micah A. Luftig.

**Data curation:** Lauren E. Haynes.

**Formal analysis:** Lauren E. Haynes.

**Funding acquisition:** Micah A. Luftig.

**Investigation:** Lauren E. Haynes, Ashley P. Barry.

**Methodology:** Lauren E. Haynes, Micah A. Luftig.

**Project administration:** Micah A. Luftig.

**Resources:** Micah A. Luftig.

**Supervision:** Micah A. Luftig.

**Validation:** Lauren E. Haynes.

**Visualization:** Lauren E. Haynes.

**Writing – original draft:** Lauren E. Haynes.

**Writing – review & editing:** Lauren E. Haynes, Ashley P. Barry, Micah A. Luftig.

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
