## [Decision Letter · Decision Letter 0]

16 Nov 2025

Single-Cell Profiling of HDAC Inhibitor-Induced EBV Lytic Heterogeneity Defines Abortive and Refractory States in B Lymphoblasts

PLOS Pathogens

Dear Dr. Luftig,

Thank you for submitting your manuscript to PLOS Pathogens. After careful consideration, we feel that it has merit but does not fully meet PLOS Pathogens's publication criteria as it currently stands. Therefore, we invite you to submit a revised version of the manuscript that addresses the points raised during the review process.

We look forward to receiving your revised manuscript.

Kind regards,

John Karijolich, Ph.D.

Academic Editor

PLOS Pathogens

Robert Kalejta

Section Editor

Editor-in-Chief

PLOS Pathogens

orcid.org/0000-0003-2946-9497

Editor-in-Chief

PLOS Pathogens

orcid.org/0000-0002-7699-2064

**Journal Requirements:**

1) Please upload all main figures as separate Figure files in .tif or .eps format. For more information about how to convert and format your figure files please see our guidelines:

2) Some material included in your submission may be copyrighted. According to PLOSu2019s copyright policy, authors who use figures or other material (e.g., graphics, clipart, maps) from another author or copyright holder must demonstrate or obtain permission to publish this material under the Creative Commons Attribution 4.0 International (CC BY 4.0) License used by PLOS journals. Please closely review the details of PLOSu2019s copyright requirements here: PLOS Licenses and Copyright. If you need to request permissions from a copyright holder, you may use PLOS's Copyright Content Permission form.

Potential Copyright Issues:

i) Figure 5. Please confirm whether you drew the images / clip-art within the figure panels by hand. If you did not draw the images, please provide (a) a link to the source of the images or icons and their license / terms of use; or (b) written permission from the copyright holder to publish the images or icons under our CC BY 4.0 license. Alternatively, you may replace the images with open source alternatives. See these open source resources you may use to replace images / clip-art:

3) When completing the data availability statement of the submission form, you indicated that you will make your data available on acceptance. We strongly recommend all authors decide on a data sharing plan before acceptance, as the process can be lengthy and hold up publication timelines. Please note that, though access restrictions are acceptable now, your entire data will need to be made freely accessible if your manuscript is accepted for publication. This policy applies to all data except where public deposition would breach compliance with the protocol approved by your research ethics board. If you are unable to adhere to our open data policy, please kindly revise your statement to explain your reasoning and we will seek the editor's input on an exemption. Please be assured that, once you have provided your new statement, the assessment of your exemption will not hold up the peer review process.

4) Please amend your detailed Financial Disclosure statement. This is published with the article. It must therefore be completed in full sentences and contain the exact wording you wish to be published.

2) If any authors received a salary from any of your funders, please state which authors and which funders..

5)  Please ensure that the funders and grant numbers match between the Financial Disclosure field and the Funding Information tab in your submission form. Note that the funders must be provided in the same order in both places as well.

**Reviewers' Comments:**

Reviewer's Responses to Questions

**Part I - Summary**

Reviewer #1: EBV causes human malignancies, and most EBV+ tumor cells contain latent EBV infection. "Lytic induction therapy" (converting EBV-infected tumor cells from the latent to lytic form of EBV infection) is being pursued as a potential method for treating latently-infected EBV+ tumors in humans, but a hurdle to using this therapy is the relatively low level of lytic infection induced by currently available drugs, including HDAC inhibitors. In this manuscript, the authors have used a variety of techniques (including single cell RNA-seq) to characterize the effects of a pan-HDAC inhibitor, panobinostat, versus a class I-specific HDAC inhibitor, nanatinostat, on the growth, survival, and lytic reactivation of four EBV-positive cell lines, including two related Burkitt lymphoma (BL) lines, (P3HR1-ZHT BL and Jijoye BL), one DLBCL line (IBL-1} and a newly infected lymphoblastoid cell line (LCL). The results of these experiments show that 1) the pan-HDAC inhibitor is more generally toxic compared to the HDAC I specific inhibitor; 2) the lytic induction efficiency of the HDAC inhibitors is much efficient in the two BL lines in comparison to the DLBCL and LCL lines; 3) even in the BL lines only a minority of the HDAC treated cells enter lytic infection, and often cells have only "abortive" lytic infection (expressing the BZLF1 immediate early gene but often not early lytic or late lytic genes); and 4) single-cell RNA-seq analysis of HDAC inhibitor treated P3HR1 BL cells identifies a "cluster 3" set of cells refractory to lytic induction that over-express a gene set consistent with NF-KB activation, including the CD137/CD137L genes. The authors suggest that the CD137/CD137L pathway may restrict lytic EBV reactivation in response to HDAC inhibitors. A strength of the manuscript is the use of several cutting edge techniques (including single cell RNA-seq and RNA hybridization probes) to identify how different EBV+ cells respond to HDAC inhibitors. Weaknesses are outlined below.

Reviewer #2: Epstein-Barr virus causes and maintains a latent reservoir in multiple malignancies making them difficult to treat. Viral reactivation using histone deacetylase inhibitors (HDACi) is being tested as a candidate therapeutic strategy yet has been met with limited success due to their inefficiency. In this manuscript, Haynes et al. explore potential mechanisms of why HDACis only lead to heterogenous responses. They phenotypically characterized the effects of class I HDACi nanatinostat and the pan-HDACi panabinostat on EBV+ lymphoblast cell lines (two Burkitt lymphoma: Jiyoye and P3HR1, one AIDS-associated immunoblastic lymphoma IBL1, and an LCL). HDACi treatment consistently led to an increase in cell cycle arrest yet had variable effects in reactivation. HDACi treatment induced lytic reactivation in BL cells but not in IBL1 and LCL. To gain insights on why majority of HDACi-treated cells are refractory to reactivation, they performed scRNAseq and compared it to their previous work. They show that abortive lytic cells have increased immune signaling genes such as STAT1/5A, CD137/L, and NFKB targets. Defining these genes that prevent lytic reactivation will inform future studies in developing more effective treatment for EBV cancers. Over-all, the study is descriptive in nature and relatively straightforward.

I only have a couple of minor comments:

1. For flow plots in Fig. 2 and 3, visualization of the BMRF1 subpopulation in the cell cycle plot and cleaved caspase subpopulation in the cell death-lytic panel are difficult to see. It would be better if they just express the subpopulations of the BMRF1 (Fig. 2) and active casp-3 (Fig. 3) as separate plots. It is unclear what the % BMRF1 is relative to in Fig. 2 --- is that from just the S phase? or total cells?

2. Can you clarify why there are no BMRF1+Zombie violet+ double positives in Fig. 3? Don't late apoptotic cells eventually lose membrane integrity? Is it because the timepoint shown is too early to see this?

3. Fig. 4. The HDACi-treated Jiyoye was compared to lytic induced P3HR1-ZHT cells. The differences in cellular background make it hard to draw robust conclusions from this comparison. Moreover, we also need to rule out that the more efficient induction of double positive cells in 4HT-treated P3HR1-ZHT cells is because ZTA is already expressed and just translocates to the nucleus (i.e. the barrier of breaking reactivation is inherently lower). A more straightforward comparison would be to examine lytic-induced Jiyoye (by Ig-crosslinking?) and HDACi-treated Jiyoye OR HDACi-treated P3HR1-ZHT with 4HT-treated or ig-crosslinked P3HR1-ZHT cells.

4. Fig. 5C. Lytic modules are not obvious. Having a zoomed inset of cluster 8 would help visualize the IE, E and L modules.

**Part II – Major Issues: Key Experiments Required for Acceptance**

Please use this section to detail the key new experiments or modifications of existing experiments that should be absolutely required to validate study conclusions.required to validate study conclusions.

Reviewer #1: 1. Although the authors suggest that the CD137/CD137L signaling pathway may be an important restriction factor preventing efficient lytic EBV reactivation in B cells, they do not include the essential experiments to support this statement. For example, does knock-down of either CD137 or CD137L increase the amount of lytic induction in response to HDAC inhibitors? Do the refractory LCL and DLBCL lines express higher levels of CD137/CD137L compared to the BL lines?

2. Almost no information is provided regarding the effects of the HDAC inhibitors on the expression of EBV latency genes in the four different cell lines. The authors must have this information at the single-cell RNA-seq level in the P3HR1 cells, although they only show results for the LMP1 latency gene. In particular they need to show whether the HDAC inhibitors increase expression of the EBV LMP2A protein in the various cell lines (preferably at the protein level), given that LMP2A is known to decrease lytic EBV infection.

3. As the lytic refractory "cluster 3" found by single cell RNA-seq in P3HR1 cells has an RNA expression pattern consistent with NF-KB (and/or LMP1) activation, and several papers have previously reported that NF-KB signaling (and LMP1) inhibits lytic EBV reactivation in EBV-infected B cells, the authors should compare the level of HDAC inhibitor induced canonical and non-canonical NF-KB proteins in the four different cell lines. Do the refractory lines have increased NF-KB activity in response to HDAC inhibitors?

Reviewer #2: None

**Part III – Minor Issues: Editorial and Data Presentation Modifications**

Reviewer #1: 1. The authors should mention that the P3HR1 "subclone" of Jijoye cells has an EBNA2 deletion and has "Wp-restricted" viral latency while the Jijoye BL cells have type III EBV latency. Thus neither cell line is a particularly good model system for studying lytic induction in human BL tumors, which generally have type I viral latency.

2. The authors should compare the cellular toxicity of the two different HDAC inhibitor drugs in EBV-infected Burkitt cells versus the same cells that have spontaneously lost the EBV genome (either EBV+ versus EBV-negative Akata BL cells or EBV+ versus EBV-negative Mutu BL cells) to determine if any of the drug induced cell killing is related to the presence of EBV infection.

Reviewer #2: See above

PLOS authors have the option to publish the peer review history of their article (what does this mean? ). If published, this will include your full peer review and any attached files.). If published, this will include your full peer review and any attached files.

**Do you want your identity to be public for this peer review?** For information about this choice, including consent withdrawal, please see our For information about this choice, including consent withdrawal, please see our Privacy Policy ..

Reviewer #1: No

Reviewer #2: No

**Figure resubmission:**

**Reproducibility:**



---

## [Decision Letter · Decision Letter 1]

9 Mar 2026

Dear Dr. Luftig,

We are pleased to inform you that your manuscript 'Single-Cell Profiling of HDAC Inhibitor-Induced EBV Lytic Heterogeneity Defines Abortive and Refractory States in B Lymphoblasts' has been provisionally accepted for publication in PLOS Pathogens.

Best regards,

John Karijolich, Ph.D.

Academic Editor

PLOS Pathogens

Robert Kalejta

Section Editor

PLOS Pathogens

Sumita Bhaduri-McIntosh

Editor-in-Chief

PLOS Pathogens

orcid.org/0000-0003-2946-9497

Michael Malim

Editor-in-Chief

PLOS Pathogens

orcid.org/0000-0002-7699-2064

Reviewer Comments (if any, and for reference):

Reviewer's Responses to Questions

**Part I - Summary**

Reviewer #1: The authors have responded adequately to the previous reviewers' comments, including performing a number of new experiments, and the resubmitted manuscript is improved.

Reviewer #2: The manuscript has significantly improved and the authors have addressed my minor concerns from the previous reviews.

I have a few more minor editorial suggestions based on the new data:

1. It would be helpful if the Review Fig. 3 (latency genes in the scRNA-seq) is included as part of Fig. S7. Reviewer 1 raised a good point and will also likely be important for readers to see.

2. RE: their new data on the CD137/137L KO, the statement “70% increase in the percentage of early lytic cells and a ~60% increase in the percentage of late lytic cells” gives an impression that their phenotypes are huge. Consider revising this to explicitly state the the total % in the CD46 vs. CD137 KO (9.55% vs. 12.8% and 1.39 vs 2.21%) since the total changes in numbers are very modest.

**Part II – Major Issues: Key Experiments Required for Acceptance**

Please use this section to detail the key new experiments or modifications of existing experiments that should be absolutely required to validate study conclusions.required to validate study conclusions.

Reviewer #1: see above comments

Reviewer #2: N/A

**Part III – Minor Issues: Editorial and Data Presentation Modifications**

Reviewer #1: (No Response)

Reviewer #2: See above

PLOS authors have the option to publish the peer review history of their article (what does this mean? ). If published, this will include your full peer review and any attached files.). If published, this will include your full peer review and any attached files.

**Do you want your identity to be public for this peer review?** For information about this choice, including consent withdrawal, please see our For information about this choice, including consent withdrawal, please see our Privacy Policy ..

Reviewer #1: No

Reviewer #2: No

---

## [Editor Report · Acceptance letter]

Dear Dr. Luftig,

We are delighted to inform you that your manuscript, "Single-Cell Profiling of HDAC Inhibitor-Induced EBV Lytic Heterogeneity Defines Abortive and Refractory States in B Lymphoblasts," has been formally accepted for publication in PLOS Pathogens.

Best regards,

Sumita Bhaduri-McIntosh

Editor-in-Chief

PLOS Pathogens

orcid.org/0000-0003-2946-9497

Michael Malim

Editor-in-Chief

PLOS Pathogens

orcid.org/0000-0002-7699-2064